# NAIS-NET: Stable Deep Networks from Non-Autonomous Differential Equations

**Marco Ciccone**[*]
Politecnico di Milano
NNAISENSE SA
marco.ciccone@polimi.it

**Marco Gallieri**[*][†]
NNAISENSE SA
marco@nnaisense.com

**Jonathan Masci**
NNAISENSE SA
jonathan@nnaisense.com

**Christian Osendorfer**
NNAISENSE SA
christian@nnaisense.com

**Faustino Gomez**
NNAISENSE SA
tino@nnaisense.com

## Abstract

This paper introduces *Non-Autonomous Input-Output Stable Network* (NAIS-Net), a very deep architecture where each stacked processing block is derived from a time-invariant non-autonomous dynamical system. Non-autonomy is implemented by skip connections from the block input to each of the unrolled processing stages and allows stability to be enforced so that blocks can be unrolled adaptively to a *pattern-dependent processing depth*. NAIS-Net induces *non-trivial, Lipschitz input-output maps*, even for an infinite unroll length. We prove that the network is globally asymptotically stable so that for every initial condition there is exactly one input-dependent equilibrium assuming $tanh$ units, and multiple stable equilibria for ReL units. An efficient implementation that enforces the stability under derived conditions for both fully-connected and convolutional layers is also presented. Experimental results show how NAIS-Net exhibits stability in practice, yielding a significant reduction in *generalization gap* compared to ResNets.

## 1   Introduction

Deep neural networks are now the state-of-the-art in a variety of challenging tasks, ranging from object recognition to natural language processing and graph analysis [28, 3, 52, 43, 36]. With enough layers, they can, in principle, learn arbitrarily complex abstract representations through an iterative process [13] where each layer transforms the output from the previous layer non-linearly until the input pattern is embedded in a latent space where inference can be done efficiently.

Until the advent of Highway [40] and Residual (ResNet; [18]) networks, training nets beyond a certain depth with gradient descent was limited by the vanishing gradient problem [19, 4]. These very deep networks (VDNNs) have skip connections that provide shortcuts for the gradient to flow back through hundreds of layers. Unfortunately, training them still requires extensive hyper-parameter tuning, and, even if there were a principled way to determine the optimal number of layers or *processing depth* for a given task, it still would be fixed for all patterns.

Recently, several researchers have started to view VDNNs from a dynamical systems perspective. Haber and Ruthotto [15] analyzed the stability of ResNets by framing them as an Euler integration of an ODE, and [34] showed how using other numerical integration methods induces various existing network architectures such as PolyNet [50], FractalNet [30] and RevNet [11]. A fundamental problem with the dynamical systems underlying these architectures is that they are *autonomous*: the input pattern sets the initial condition, only directly affecting the first processing stage. This means that if

---

[*]The authors equally contributed.
[†]The author derived the mathematical results.

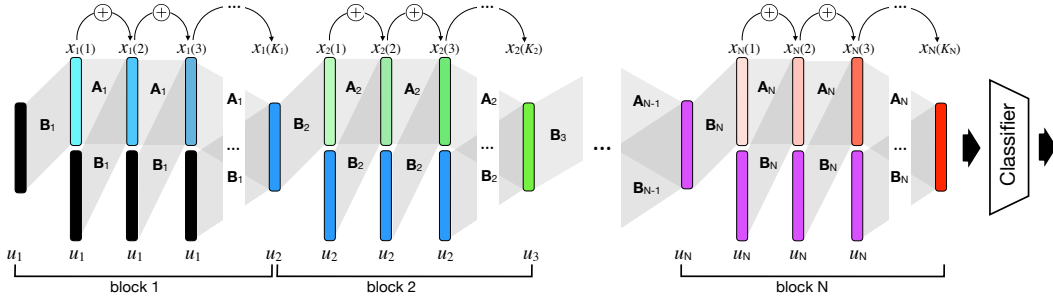

Figure 1: **NAIS-Net architecture**. Each block represents a time-invariant iterative process as the first layer in the $i$-th block, $x_i(1)$, is unrolled into a pattern-dependent number, $K_i$, of processing stages, using weight matrices $\mathbf{A}_i$ and $\mathbf{B}_i$. The skip connections from the input, $u_i$, to all layers in block $i$ make the process non-autonomous. Blocks can be chained together (each block modeling a different latent space) by passing final latent representation, $x_i(K_i)$, of block $i$ as the input to block $i + 1$.

the system converges, there is either exactly one fixpoint or exactly one limit cycle [42]. Neither case is desirable from a learning perspective because a dynamical system should have input-dependent convergence properties so that representations are useful for learning. One possible approach to achieve this is to have a *non-autonomous* system where, at each iteration, the system is forced by an external input.

This paper introduces a novel network architecture, called the *"Non-Autonomous Input-Output Stable Network"* (NAIS-Net), that is derived from a dynamical system that is both time-invariant (weights are shared) and non-autonomous.[3] NAIS-Net is a general residual architecture where a block (see figure 1) is the unrolling of a time-invariant system, and non-autonomy is implemented by having the external input applied to each of the unrolled processing stages in the block through skip connections. ResNets are similar to NAIS-Net except that ResNets are time-varying and only receive the external input at the first layer of the block.

With this design, we can derive sufficient conditions under which the network exhibits input-dependent equilibria that are globally asymptotically stable for every initial condition. More specifically, in section 3, we prove that with $tanh$ activations, NAIS-Net has exactly one input-dependent equilibrium, while with ReLU activations it has multiple stable equilibria per input pattern. Moreover, the NAIS-Net architecture allows not only the internal stability of the system to be analyzed but, more importantly, the input-output stability — the difference between the representations generated by two different inputs belonging to a bounded set will also be bounded at each stage of the unrolling.[4]

In section 4, we provide an efficient implementation that enforces the stability conditions for both fully-connected and convolutional layers in the stochastic optimization setting. These implementations are compared experimentally with ResNets on both CIFAR-10 and CIFAR-100 datasets, in section 5, showing that NAIS-Nets achieve comparable classification accuracy with a much better *generalization gap*. NAIS-Nets can also be 10 to 20 times deeper than the original ResNet without increasing the total number of network parameters, and, by stacking several stable NAIS-Net blocks, models that implement pattern-dependent processing depth can be trained without requiring any normalization at each step (except when there is a change in layer dimensionality, to speed up training).

The next section presents a more formal treatment of the dynamical systems perspective of neural networks, and a brief overview of work to date in this area.

## 2 Background and Related Work

Representation learning is about finding a mapping from input patterns to encodings that disentangle the underlying variational factors of the input set. With such an encoding, a large portion of typical supervised learning tasks (e.g. classification and regression) should be solvable using just a simple model like logistic regression. A key characteristic of such a mapping is its invariance to input transformations that do not alter these factors for a given input[5]. In particular, random perturbations of the input should in general not be drastically amplified in the encoding. In the field of control

theory, this property is central to stability analysis which investigates the properties of dynamical systems under which they converge to a single steady state without exhibiting chaos [25, 42, 39].

In machine learning, stability has long been central to the study of recurrent neural networks (RNNs) with respect to the vanishing [19, 4, 37], and exploding [9, 2, 37] gradient problems, leading to the development of Long Short-Term Memory [20] to alleviate the former. More recently, general conditions for RNN stability have been presented [52, 24, 31, 47] based on general insights related to Matrix Norm analysis. Input-output stability [25] has also been analyzed for simple RNNs [41, 26, 16, 38].

Recently, the stability of deep feed-forward networks was more closely investigated, mostly due to adversarial attacks [44] on trained networks. It turns out that sensitivity to (adversarial) input perturbations in the inference process can be avoided by ensuring certain conditions on the spectral norms of the weight matrices [7, 49]. Additionally, special properties of the spectral norm of weight matrices mitigate instabilities during the training of Generative Adversarial Networks [35].

Almost all successfully trained VDNNs [20, 18, 40, 6] share the following core building block:

$$x(k + 1) = x(k) + f\left(x(k), \theta(k)\right), 1 \le k \le K. \tag{1}$$

That is, in order to compute a vector representation at layer $k + 1$ (or time $k + 1$ for recurrent networks), *additively update* $x(k)$ with some non-linear transformation $f(\cdot)$ of $x(k)$ which depends on parameters $\theta(k)$. The reason usual given for why Eq. (1) allows VDNNs to be trained is that the explicit identity connections avoid the vanishing gradient problem.

The semantics of the forward path are however still considered unclear. A recent interpretation is that these feed-forward architectures implement *iterative inference* [13, 23]. This view is reinforced by observing that Eq. (1) is a forward Euler discretization [1] of the ordinary differential equation (ODE) $\dot{x}(t) = f(x(t), \Theta)$ if $\theta(k) \equiv \Theta$ for all $1 \le k \le K$ in Eq. (1). This connection between dynamical systems and feed-forward architectures was recently also observed by several other authors [48]. This point of view leads to a large family of new network architectures that are induced by various numerical integration methods [34]. Moreover, stability problems in both the forward as well the backward path of VDNNs have been addressed by relying on well-known analytical approaches for continuous-time ODEs [15, 5]. In the present paper, we instead address the problem directly in discrete-time, meaning that our stability result is preserved by the network implementation. With the exception of [33], none of this prior research considers time-invariant, non-autonomous systems.

Conceptually, our work shares similarities with approaches that build network according to iterative algorithms [14, 51] and recent ideas investigating pattern-dependent processing time [12, 46, 10].

## 3 Non-Autonomous Input-Output Stable Nets (NAIS-Nets)

This section provides stability conditions for both fully-connected and convolutional NAIS-Net layers. We formally prove that NAIS-Net provides a non-trivial input-dependent output for each iteration $k$ as well as in the asymptotic case ($k \to \infty$). The following dynamical system:

$$x(k + 1) = x(k) + hf\left(x(k), u, \theta\right), \; x(0) = 0, \tag{2}$$

is used throughout the paper, where $x \in \mathbb{R}^n$ is the latent state, $u \in \mathbb{R}^m$ is the network input, and $h > 0$. For ease of notation, in the remainder of the paper the explicit dependence on the parameters, $\theta$, will be omitted.

**Fully Connected NAIS-Net Layer.**  Our fully connected layer is defined by

$$x(k + 1) = x(k) + h\sigma\left(Ax(k) + Bu + b\right), \tag{3}$$

where $A \in \mathbb{R}^{n \times n}$ and $B \in \mathbb{R}^{n \times m}$ are the state and input transfer matrices, and $b \in \mathbb{R}^n$ is a bias. The activation $\sigma \in \mathbb{R}^n$ is a vector of (element-wise) instances of an activation function, denoted as $\sigma_i$ with $i \in \{1, \dots, n\}$. In this paper, we only consider the hyperbolic tangent, $tanh$, and Rectified Linear Units (ReLU) activation functions. Note that by setting $B = 0$, and the step $h = 1$ the original ResNet formulation is obtained.

**Convolutional NAIS-Net Layer.**  The architecture can be easily extended to Convolutional Networks by replacing the matrix multiplications in Eq. (3) with a convolution operator:

$$X(k + 1) = X(k) + h\sigma\left(C * X + D * U + E\right). \tag{4}$$

Consider the case of $N_C$ channels. The convolutional layer in Eq. (4) can be rewritten, for each latent map $c \in \{1, 2, \ldots, N_C\}$, in the equivalent form:

$$X^c(k+1) = X^c(k) + h\sigma \left( \sum_i^{N_C} C_i^c * X^i(k) + \sum_j^{N_C} D_j^c * U^j + E^c \right),$$ (5)

where: $X^i(k) \in \mathbb{R}^{n_X \times n_X}$ is the layer state matrix for channel $i$, $U^j \in \mathbb{R}^{n_U \times n_U}$ is the layer input data matrix for channel $j$ (where an appropriate zero padding has been applied) at layer $k$, $C_i^c \in \mathbb{R}^{n_C \times n_C}$ is the state convolution filter from state channel $i$ to state channel $c$, $D_j^c$ is its equivalent for the input, and $E^c$ is a bias. The activation, $\sigma$, is still applied element-wise. The convolution for $X$ has a fixed stride $s = 1$, a filter size $n_C$ and a zero padding of $p \in \mathbb{N}$, such that $n_C = 2p + 1$.[6]

Convolutional layers can be rewritten in the same form as fully connected layers (see proof of Lemma 1 in the supplementary material). Therefore, the stability results in the next section will be formulated for the fully connected case, but apply to both.

**Stability Analysis.**  Here, the stability conditions for NAIS-Nets which were instrumental to their design are laid out. We are interested in using a cascade of unrolled NAIS blocks (see Figure 1), where each block is described by either Eq. (3) or Eq. (4). Since we are dealing with a cascade of dynamical systems, then stability of the entire network can be enforced by having stable blocks [25].

The state-transfer Jacobian for layer $k$ is defined as:

$$J(x(k), u) = \frac{\partial x(k+1)}{\partial x(k)} = I + h\frac{\partial \sigma(\Delta x(k))}{\partial \Delta x(k)} A,$$ (6)

where the argument of the activation function, $\sigma$, is denoted as $\Delta x(k)$. Take an arbitrarily small scalar $\underline{\sigma} > 0$ and define the set of pairs $(x, u)$ for which the activations are not saturated as:

$$\mathcal{P} = \left\{ (x, u) : \frac{\partial \sigma_i(\Delta x(k))}{\partial \Delta x_i(k)} \geq \underline{\sigma}, \ \forall i \in [1, 2, \ldots, n] \right\}.$$ (7)

Theorem 1 below proves that the non-autonomuous residual network produces a bounded output given a bounded, possibly noisy, input, and that the network state converges to a constant value as the number of layers tends to infinity, if the following stability condition holds:

**Condition 1.** For any $\underline{\sigma} > 0$, the Jacobian satisfies:

$$\bar{\rho} = \sup_{(x,u)\in\mathcal{P}} \rho(J(x, u)), \ \text{s.t.} \ \bar{\rho} < 1,$$ (8)

where $\rho(\cdot)$ is the spectral radius.

The steady states, $\bar{x}$, are determined by a *continuous* function of $u$. This means that a small change in $u$ cannot result in a very different $\bar{x}$. For $tanh$ activation, $\bar{x}$ depends linearly on $u$, therefore the block needs to be unrolled for a finite number of iterations, $K$, for the mapping to be non-linear. That is not the case for ReLU, which can be unrolled indefinitely and still provide a piece-wise affine mapping.

In Theorem 1, the Input-Output (IO) gain function, $\gamma(\cdot)$, describes the effect of norm-bounded input perturbations on the network trajectory. This gain provides insight as to the level of robust invariance of the classification regions to changes in the input data with respect to the training set. In particular, as the gain is decreased, the perturbed solution will be *closer* to the solution obtained from the training set. This can lead to increased robustness and generalization with respect to a network that does not statisfy Condition 1. Note that the IO gain, $\gamma(\cdot)$, is linear, and hence the block IO map is *Lipschitz* even for an *infinite* unroll length. The IO gain depends directly on the norm of the state transfer Jacobian, in Eq. (8), as indicated by the term $\bar{\rho}$ in Theorem 1.[7]

**Theorem 1.** (Asymptotic stability for shared weights)
If Condition 1 holds, then NAIS-Net with ReLU or $tanh$ activations is Asymptotically Stable with respect to *input dependent* equilibrium points. More formally:

$$x(k) \to \bar{x} \in \mathbb{R}^n, \ \forall x(0) \in \mathcal{X} \subseteq \mathbb{R}^n, \ u \in \mathbb{R}^m.$$ (9)

The trajectory is described by $\|x(k) - \bar{x}\| \leq \bar{\rho}^k \|x(0) - \bar{x}\|$, where $\|\cdot\|$ is a suitable matrix norm.

| **Algorithm 1** Fully Connected Reprojection | **Algorithm 2** CNN Reprojection |
|---|---|

**Algorithm 1** Fully Connected Reprojection

**Input:** $R \in \mathbb{R}^{\tilde{n} \times n}$, $\tilde{n} \le n$, $\delta = 1 - 2\epsilon$, $\epsilon \in (0, 0.5)$.

**if** $\|R^T R\|_F > \delta$ **then**

$\quad \tilde{R} \leftarrow \sqrt{\delta} \frac{R}{\sqrt{\|R^T R\|_F}}$

**else**
$\quad \tilde{R} \leftarrow R$
**end if**
**Output:** $\tilde{R}$

**Algorithm 2** CNN Reprojection

**Input:** $\delta \in \mathbb{R}^{N_C}$, $C \in \mathbb{R}^{n_X \times n_X \times N_C \times N_C}$, and $0 < \epsilon < \eta < 1$.

**for** each feature map $c$ **do**

$\quad \tilde{\delta}_c \leftarrow \max \left( \min \left( \delta_c, 1 - \eta \right), -1 + \eta \right)$

$\quad \tilde{C}^c_{i_{\text{centre}}} \leftarrow -1 - \tilde{\delta}_c$

$\quad$ **if** $\sum_{j \neq i_{\text{centre}}} |C^c_j| > 1 - \epsilon - |\tilde{\delta}_c|$ **then**

$\quad\quad \tilde{C}^c_j \leftarrow \left( 1 - \epsilon - |\tilde{\delta}_c| \right) \frac{C^c_j}{\sum_{j \neq i_{\text{centre}}} |C^c_j|}$

$\quad$ **end if**
**end for**
**Output:** $\tilde{\delta}, \tilde{C}$

Figure 2: Proposed algorithms for enforcing stability.

In particular:

- With $tanh$ activation, the steady state $\bar{x}$ is independent of the initial state, and it is a linear function of the input, namely, $\bar{x} = A^{-1} B u$. The network is Globally Asymptotically Stable.

  With ReLU activation, $\bar{x}$ is given by a continuous piecewise affine function of $x(0)$ and $u$. The network is Locally Asymptotically Stable with respect to each $\bar{x}$.

- If the activation is $tanh$, then the network is Globally Input-Output (robustly) Stable for any additive input perturbation $w \in \mathbb{R}^m$. The trajectory is described by:

$$\|x(k) - \bar{x}\| \le \bar{\rho}^k \|x(0) - \bar{x}\| + \gamma(\|w\|), \text{ with } \gamma(\|w\|) = h \frac{\|B\|}{(1 - \bar{\rho})} \|w\|. \qquad (10)$$

  where $\gamma(\cdot)$ is the input-output gain. For any $\mu \ge 0$, if $\|w\| \le \mu$ then the following set is robustly positively invariant ($x(k) \in \mathcal{X}, \forall k \ge 0$):

$$\mathcal{X} = \{ x \in \mathbb{R}^n : \|x - \bar{x}\| \le \gamma(\mu) \}. \qquad (11)$$

- If the activation is ReLU, then the network is Globally Input-Output practically Stable. In other words, $\forall k \ge 0$ we have:

$$\|x(k) - \bar{x}\| \le \bar{\rho}^k \|x(0) - \bar{x}\| + \gamma(\|w\|) + \zeta. \qquad (12)$$

  The constant $\zeta \ge 0$ is the norm ball radius for $x(0) - \bar{x}$.

## 4 Implementation

In general, an optimization problem with a spectral radius constraint as in Eq. (8) is hard [24]. One possible approach is to relax the constraint to a singular value constraint [24] which is applicable to both fully connected as well as convolutional layer types [49]. However, this approach is only applicable if the identity matrix in the Jacobian (Eq. (6)) is scaled by a factor $0 < c < 1$ [24]. In this work we instead fulfil the spectral radius constraint directly.

The basic intuition for the presented algorithms is the fact that for a simple Jacobian of the form $I + M$, $M \in \mathbb{R}^{n \times n}$, Condition 1 is fulfilled, if $M$ has eigenvalues with real part in $(-2, 0)$ and imaginary part in the unit circle. In the supplemental material we prove that the following algorithms fulfill Condition 1 following this intuition. Note that, in the following, the presented procedures are to be performed for each block of the network.

**Fully-connected blocks.** In the fully connected case, we restrict the matrix $A$ to by symmetric and negative definite by choosing the following parameterization for them:

$$A = -R^T R - \epsilon I, \qquad (13)$$

where $R \in \mathbb{R}^{n \times n}$ is trained, and $0 < \epsilon \ll 1$ is a hyper-parameter. Then, we propose a bound on the Frobenius norm, $\|R^T R\|_F$. Algorithm 1, performed during training, implements the following[8]:

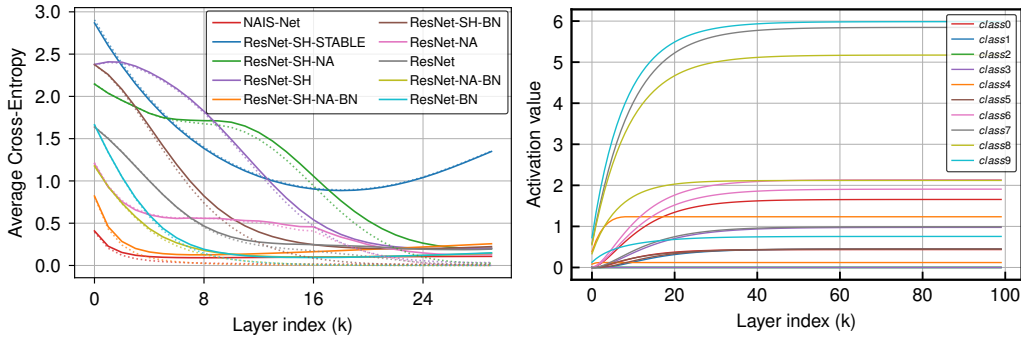

Figure 3: **Single neuron trajectory and convergence. (Left)** Average loss of NAIS-Net with different residual architectures over the unroll length. Note that both RESNET-SH-STABLE and NAIS-Net satisfy the stability conditions for convergence, but only NAIS-Net is able to learn, showing the importance of non-autonomy. **Cross-entropy loss vs processing depth. (Right)** Activation of a NAIS-Net single neuron for input samples from each class on MNIST. Trajectories not only differ with respect to the actual steady-state but also with respect to the convergence time.

**Theorem 2.** (Fully-connected weight projection)
Given $R \in \mathbb{R}^{n \times n}$, the projection $\tilde{R} = \sqrt{\delta} \frac{R}{\sqrt{\|R^T R\|_F}}$, with $\delta = 1 - 2\epsilon \in (0, 1)$, ensures that $A = -\tilde{R}^T \tilde{R} - \epsilon I$ is such that Condition 1 is satisfied for $h \leq 1$ and therefore Theorem 1 holds.

Note that $\delta = 2(1 - \epsilon) \in (0, 2)$ is also sufficient for stability, however, the $\delta$ from Theorem 2 makes the trajectory free from oscillations (critically damped), see Figure 3. This is further discussed in Appendix.

**Convolutional blocks.** The symmetric parametrization assumed in the fully connected case can not be used for a convolutional layer. We will instead make use of the following result:

**Lemma 1.** The convolutional layer Eq. (4) with zero-padding $p \in \mathbb{N}$, and filter size $n_C = 2p + 1$ has a Jacobian of the form Eq. (6). with $A \in \mathbb{R}^{n_X^2 N_C \times n_X^2 N_C}$. The diagonal elements of this matrix, namely, $A_{n_X^2 c + j, n_X^2 c + j}$, $0 \leq c < N_C$, $0 \leq j < n_X^2$ are the central elements of the $(c + 1)$-th convolutional filter mapping $X^{c+1}(k)$, into $X^{c+1}(k + 1)$, denoted by $C_{i_{\text{centre}}}^c$. The other elements in row $n_X^2 c + j$, $0 \leq c < N_C$, $0 \leq j < n_X^2$ are the remaining filter values mapping to $X^{(c+1)}(k + 1)$.

To fulfill the stability condition, the first step is to set $C_{i_{\text{centre}}}^c = -1 - \delta_c$, where $\delta_c$ is trainable parameter satisfying $|\delta_c| < 1 - \eta$, and $0 < \eta \ll 1$ is a hyper-parameter. Then we will suitably bound the $\infty$-norm of the Jacobian by constraining the remaining filter elements. The steps are summarized in Algorithm 2 which is inspired by the Gershgorin Theorem [21]. The following result is obtained:

**Theorem 3.** (Convolutional weight projection)
Algorithm 2 fulfils Condition 1 for the convolutional layer, for $h \leq 1$, hence Theorem 1 holds.

Note that the algorithm complexity scales with the number of filters. A simple design choice for the layer is to set $\delta = 0$, which results in $C_{i_{\text{centre}}}^c$ being fixed at $-1$[9].

# 5 Experiments

Experiments were conducted comparing NAIS-Net with ResNet, and variants thereof, using both fully-connected (MNIST, section 5.1) and convolutional (CIFAR-10/100, section 5.2) architectures to quantitatively assess the performance advantage of having a VDNN where stability is enforced.

## 5.1 Preliminary Analysis on MNIST

For the MNIST dataset [32] a single-block NAIS-Net was compared with 9 different 30-layer ResNet variants each with a different combination of the following features: **SH** (shared weights i.e. time-invariant), **NA** (non-autonomous i.e. input skip connections), **BN** (with Batch Normalization), **Stable**

(stability enforced by Algorithm 1). For example, RESNET-SH-NA-BN refers to a 30-layer ResNet that is time-invariant because weights are shared across all layers (SH), non-autonomous because it has skip connections from the input to all layers (NA), and uses batch normalization (BN). Since NAIS-Net is time-invariant, non-autonomous, and input/output stable (i.e. SH-NA-STABLE), the chosen ResNet variants represent ablations of the these three features. For instance, RESNET-SH-NA is a NAIS-Net without I/O stability being enforced by the reprojection step described in Algorithm 1, and RESNET-NA, is a non-stable NAIS-Net that is time-variant, i.e non-shared-weights, etc. The NAIS-Net was unrolled for $K = 30$ iterations for all input patterns. All networks were trained using stochastic gradient descent with momentum $0.9$ and learning rate $0.1$, for 150 epochs.

**Results.** Test accuracy for NAIS-NET was $97.28\%$, while RESNET-SH-BN was second best with $96.69\%$, but without BatchNorm (RESNET-SH) it only achieved $95.86\%$ (averaged over 10 runs).

After training, the behavior of each network variant was analyzed by passing the activation, $x(i)$, though the softmax classifier and measuring the cross-entropy loss. The loss at each iteration describes the trajectory of each sample in the latent space: the closer the sample to the correct steady state the closer the loss to zero (see Figure 3). All variants initially refine their predictions at each iteration since the loss tends to decreases at each layer, but at different rates. However, NAIS-Net is the only one that does so monotonically, not increasing loss as $i$ approaches 30. Figure 3 shows how neuron activations in NAIS-Net converge to different steady state activations for different input patterns instead of all converging to zero as is the case with RESNET-SH-STABLE, confirming the results of [15]. Importantly, NAIS-Net is able to learn even with the stability constraint, showing that non-autonomy is key to obtaining representations that are stable *and* good for learning the task.

NAIS-Net also allows training of unbounded processing depth without any feature normalization steps. Note that BN actually speeds up loss convergence, especially for RESNET-SH-NA-BN (i.e. unstable NAIS-Net). Adding BN makes the behavior very similar to NAIS-Net because BN also implicitly normalizes the Jacobian, but it does not ensure that its eigenvalues are in the stability region.

## 5.2 Image Classification on CIFAR-10/100

Experiments on image classification were performed on standard image recognition benchmarks CIFAR-10 and CIFAR-100 [27]. These benchmarks are simple enough to allow for multiple runs to test for statistical significance, yet sufficiently complex to require convolutional layers.

**Setup.** The following standard architecture was used to compare NAIS-Net with ResNet[10]: three sets of 18 residual blocks with 16, 32, and 64 filters, respectively, for a total of 54 stacked blocks. NAIS-Net was tested in two versions: NAIS-NET1 where each block is unrolled just once, for a total processing depth of 108, and NAIS-NET10 where each block is unrolled 10 times per block, for a total processing depth of 540. The initial learning rate of $0.1$ was decreased by a factor of 10 at epochs 150, 250 and 350 and the experiment were run for 450 epochs. Note that each block in the ResNet of [17] has two convolutions (plus BatchNorm and ReLU) whereas NAIS-Net unrolls with a single convolution. Therefore, to make the comparison of the two architectures as fair as possible by using the same number of parameters, a single convolution was also used for ResNet.

**Results.** Table 5.2 compares the performance on the two datasets, averaged over 5 runs. For CIFAR-10, NAIS-Net and ResNet performed similarly, and unrolling NAIS-Net for more than one iteration had little affect. This was not the case for CIFAR-100 where NAIS-NET10 improves over NAIS-NET1 by $1\%$. Moreover, although mean accuracy is slightly lower than ResNet, the variance is considerably lower. Figure 4 shows that NAIS-Net is less prone to overfitting than a classic ResNet, reducing the generalization gap by $33\%$. This is a consequence of the stability constraint which imparts a degree of robust invariance to input perturbations (see Section 3). It is also important to note that NAIS-Net can unroll up to 540 layers, and still train without any problems.

## 5.3 Pattern-Dependent Processing Depth

For simplicity, the number of unrolling steps per block in the previous experiments was fixed. A more general and potentially more powerful setup is to have the processing depth adapt automatically. Since NAIS-Net blocks are guaranteed to converge to a pattern-dependent steady state after an indeterminate number of iterations, processing depth can be controlled dynamically by terminating the unrolling process whenever the distance between a layer representation, $x(i)$, and that of the

| MODEL | CIFAR-10 TRAIN/TEST | CIFAR-100 TRAIN/TEST |
|---|---|---|
| RESNET | 99.86±0.03 | 97.42 ± 0.06 |
| | 91.72±0.38 | 66.34 ± 0.82 |
| NAIS-NET1 | 99.37±0.08 | 86.90 ± 1.47 |
| | 91.24±0.10 | 65.00 ± 0.52 |
| NAIS-NET10 | 99.50±0.02 | 86.91 ± 0.42 |
| | 91.25±0.46 | 66.07 ± 0.24 |

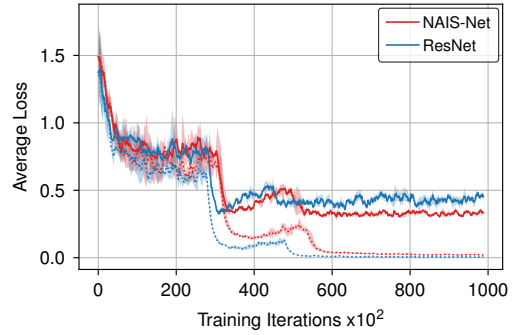

Figure 4: **CIFAR Results. (Left)** Classification accuracy on the CIFAR-10 and CIFAR-100 datasets averaged over 5 runs. **Generalization gap on CIFAR-10. (Right)** Dotted curves (training set) are very similar for the two networks but NAIS-Net has a considerably lower test curve (solid).

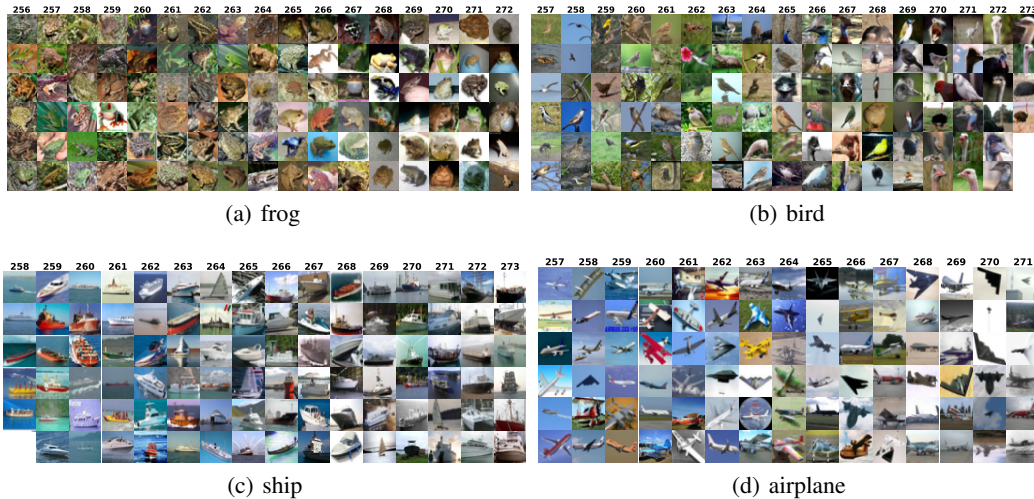

(a) frog

(b) bird

(c) ship

(d) airplane

Figure 5: **Image samples with corresponding NAIS-Net depth.** The figure shows samples from CIFAR-10 grouped by final network depth, for four different classes. The qualitative differences evident in images inducing different final depths indicate that NAIS-Net adapts processing systematically according characteristics of the data. For example, *"frog"* images with textured background are processed with fewer iterations than those with plain background. Similarly, *"ship"* and *"airplane"* images having a predominantly blue color are processed with lower depth than those that are grey/white, and *"bird"* images are grouped roughly according to bird size with larger species such as ostriches and turkeys being classified with greater processing depth. A higher definition version of the figure is made available in the supplementary materials.

immediately previous layer, $x(i-1)$, drops below a specified threshold. With this mechanism, NAIS-Net can determine the processing depth for each input pattern. Intuitively, one could speculate that similar input patterns would require similar processing depth in order to be mapped to the same region in latent space. To explore this hypothesis, NAIS-Net was trained on CIFAR-10 with an unrolling threshold of $\epsilon = 10^{-4}$. At test time the network was unrolled using the same threshold.

Figure 5 shows selected images from four different classes organized according to the final network depth used to classify them after training. The qualitative differences seen from low to high depth suggests that NAIS-Net is using processing depth as an additional degree of freedom so that, for a given training run, the network learns to use models of different complexity (depth) for different types of inputs within each class. To be clear, the hypothesis is not that depth correlates to some notion of input complexity where the same images are always classified at the same depth across runs.

## 6 Conclusions

We presented NAIS-Net, a non-autonomous residual architecture that can be unrolled until the latent space representation converges to a stable input-dependent state. This is achieved thanks to stability

and non-autonomy properties. We derived stability conditions for the model and proposed two efficient reprojection algorithms, both for fully-connected and convolutional layers, to enforce the network parameters to stay within the set of feasible solutions during training.

NAIS-Net achieves asymptotic stability and, as consequence of that, input-output stability. Stability makes the model more robust and we observe a reduction of the generalization gap by quite some margin, without negatively impacting performance. The question of scalability to benchmarks such as ImageNet [8] will be a main topic of future work.

We believe that cross-breeding machine learning and control theory will open up many new interesting avenues for research, and that more robust and stable variants of commonly used neural networks, both feed-forward and recurrent, will be possible.

## Aknowledgements

We want to thank Wojciech Jaśkowski, Rupesh Srivastava and the anonymous reviewers for their comments on the idea and initial drafts of the paper.

## Footnotes

[3]The DenseNet architecture [29, 22] is non-autonomous, but time-varying.

[4]In the supplementary material, we also show that these results hold both for shared and unshared weights.

[5]Such invariance conditions can be very powerful inductive biases on their own: For example, requiring invariance to time transformations in the input leads to popular RNN architectures [45].

[6] If $s \geq 0$, then $x$ can be extended with an appropriate number of constant zeros (not connected).

[7] see supplementary material for additional details and all proofs, where the untied case is also covered.

[8]The more relaxed condition $\delta \in (0, 2)$ is sufficient for Theorem 1 to hold locally (supplementary material).

[9]Setting $\delta = 0$ removes the need for hyper-parameter $\eta$ but does not necessarily reduce conservativeness as it will further constrain the remaining element of the filter bank. This is further discussed in the supplementary.

[10]`https://github.com/tensorflow/models/tree/master/official/resnet`

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
