[Supplementary Material]

# NAIS-NET: Stable Deep Networks from Non-Autonomous Differential Equations
## *Supplementary Material*

**Marco Ciccone**[*]
Politecnico di Milano
NNAISENSE SA
marco.ciccone@polimi.it

**Marco Gallieri**[*][†]
NNAISENSE SA
marco@nnaisense.com

**Jonathan Masci**
NNAISENSE SA
jonathan@nnaisense.com

**Christian Osendorfer**
NNAISENSE SA
christian@nnaisense.com

**Faustino Gomez**
NNAISENSE SA
tino@nnaisense.com

## A  Basic Definitions for the Tied Weight Case

Recall, from the main paper, that the stability of a NAIS-Net block with fully connected or convolutional architecture can be analyzed by means of the following vectorised representation:

$$x(k + 1) = f(x(k), u) = x(k) + h\sigma\bigg( Ax(k) + Bu + b \bigg), \tag{1}$$

where $k$ is the unroll index for the considered block. Since the blocks are cascaded, stability of each block implies stability of the full network. Hence, this supplementary material focuses on theoretical results for a single block.

### A.1  Relevant Sets and Operators

#### A.1.1  Notation

Denote the slope of the activation function vector, $\sigma(\Delta x(k))$, as the diagonal matrix, $\sigma^{'}(\Delta x(k))$, with entries:

$$\sigma^{'}_{ii}(\Delta x(k)) = \frac{\partial \sigma_i(\Delta x(k))}{\partial \Delta x_i(k)}. \tag{2}$$

The following definitions will be use to obtain the stability results, where $0 < \underline{\sigma} \ll 1$:

$$
\begin{aligned}
\mathcal{P}_i &= \{(x, u) \in \mathbb{R}^n \times \mathbb{R}^m : \sigma^{'}_{ii}(x, u) \geq \underline{\sigma}\}, \\
\mathcal{P} &= \{(x, u) \in \mathbb{R}^n \times \mathbb{R}^m : \sigma^{'}_{ii}(x, u) \geq \underline{\sigma}, \forall i\}, \\
\mathcal{N}_i &= \mathcal{P} \cup \{(x, u) \in \mathbb{R}^n \times \mathbb{R}^m : \sigma^{'}_{ii}(x, u) \in [0, \underline{\sigma})\}, \\
\mathcal{N} &= \{(x, u) \in \mathbb{R}^n \times \mathbb{R}^m : \sigma^{'}_{ii}(x, u) \in [0, \underline{\sigma}), \forall i\},
\end{aligned}
\tag{3}
$$

In particular, the set $\mathcal{P}$ is such that the activation function is not saturated as its derivative has a non-zero lower bound.

---

[*]The authors equally contributed.

[†]The author derived the mathematical results.

### A.1.2 Linear Algebra Elements

The notation, $\| \cdot \|$ is used to denote a *suitable* matrix norm. This norm will be characterized specifically on a case by case basis. The same norm will be used consistently throughout definitions, assumptions and proofs.

We will often use the following:

**Lemma 1.** (Eigenvalue shift)

Consider two matrices $A \in \mathbb{C}^{n \times n}$, and $C = cI + A$ with $c$ being a complex scalar. If $\lambda$ is an eigenvalue of $A$ then $c + \lambda$ is an eigenvalue of $C$.

*Proof.* Given any eigenvalues $\lambda$ of $A$ with corresponding eigenvector $v$ we have that:

$$\lambda v = Av \Leftrightarrow (\lambda + c)v = \lambda v + cv = Av + cv = Av + cIv = (A + cI)v = Cv \tag{4}$$

$\square$

Throughout the material, the notation $A_{i\bullet}$ is used to denote the $i$-th row of a matrix $A$.

### A.1.3 Non-autonomuous Behaviour Set

The following set will be consider throughout the paper:

**Definition A.1.** (Non-autonomous behaviour set) The set $\mathcal{P}$ is referred to as the set of *fully non-autonomous* behaviour in the extended state-input space, and its set-projection over $x$, namely,

$$\pi_x(\mathcal{P}) = \{x \in \mathbb{R}^n : \exists u \in \mathbb{R}^m, \ (x, u) \in \mathcal{P}\}, \tag{5}$$

is the set of fully non-autonomous behaviour in the state space. This is the only set in which every output dimension of the ResNet with input skip connection can be directly influenced by the input, given a non-zero[3] matrix $B$.

Note that, for a $\tanh$ activation, then we simply have that $\mathcal{P} \subseteq \mathbb{R}^{n+m}$ (with $\mathcal{P} \to \mathbb{R}^{n+m}$ for $\bar{\sigma}' \to 0$). For a ReLU activation, on the other hand, for each layer $k$ we have:

$$\mathcal{P} = \mathcal{P}(k) = \{(x, u) \in \mathbb{R}^n \times \mathbb{R}^m : \ A(k)x + B(k)u + b(k) > 0\}. \tag{6}$$

## A.2 Stability Definitions for Tied Weights

This section provides a summary of definitions borrowed from control theory that are used to describe and derive our main result. The following definitions have been adapted from [2] and refer to the general dynamical system:

$$x^+ = f(x, u). \tag{7}$$

Since we are dealing with a cascade of dynamical systems (see Figure 1 in (*main paper*)), then stability of the entire network can be enforced by having stable blocks [4]. In the remainder of this material, we will therefore address a single unroll. We will cover both the tied and untied weight case, starting from the latter as it is the most general.

### A.2.1 Describing Functions

The following functions are instrumental to describe the desired behaviour of the network output at each layer or time step.

**Definition A.2.** ($\mathcal{K}$-function) A continuous function $\alpha : \mathbb{R}_{\geq 0} \to \mathbb{R}_{\geq 0}$ is said to be a $\mathcal{K}$-function ($\alpha \in \mathcal{K}$) if it is strictly increasing, with $\alpha(0) = 0$.

**Definition A.3.** ($\mathcal{K}_\infty$-function) A continuous function $\alpha : \mathbb{R}_{\geq 0} \to \mathbb{R}_{\geq 0}$ is said to be a $\mathcal{K}_\infty$-function ($\alpha \in \mathcal{K}_\infty$) if it is a $\mathcal{K}$-function and if it is radially unbounded, that is $\alpha(r) \to \infty$ as $r \to \infty$.

**Definition A.4.** ($\mathcal{KL}$-function) A continuous function $\beta : \mathbb{R}^2_{\geq 0} \to \mathbb{R}_{\geq 0}$ is said to be a $\mathcal{KL}$-function ($\beta \in \mathcal{KL}$) if it is a $\mathcal{K}$-function in its first argument, it is positive definite and non- increasing in the second argument, and if $\beta(r, t) \to 0$ as $t \to \infty$.

The following definitions are given for time-invariant RNNs, namely DNN with tied weights. They can also be generalised to the case of untied weights DNN and time-varying RNNs by considering worst case conditions over the layer (time) index $k$. In this case the properties are said to hold *uniformly* for all $k \geq 0$. This is done in Section B. The tied-weight case follows.

### A.2.2 Invariance, Stability and Robustness

**Definition A.5.** (Positively Invariant Set) A set $\mathcal{X} \subseteq \mathbb{R}^n$ is said to be positively invariant (PI) for a dynamical system under an input $u \in \mathcal{U} \subseteq \mathbb{R}^n$ if

$$f(x, u) \in \mathcal{X}, \ \forall x \in \mathcal{X}. \tag{8}$$

**Definition A.6.** (Robustly Positively Invariant Set) The set $\mathcal{X} \subseteq \mathbb{R}^n$ is said to be *robustly positively invariant* (RPI) to additive input perturbations $w \in \mathcal{W}$ if $\mathcal{X}$ is PI for any input $\tilde{u} = u + w$, $u \in \mathcal{U}, \forall w \in \mathcal{W}$.

**Definition A.7.** (Asymptotic Stability) The system Eq. (7) is called Globally Asymptotically Stable around its equilibrium point $\bar{x}$ if it satisfies the following two conditions:

1. **Stability.** Given any $\epsilon > 0$, $\exists \delta_1 > 0$ such that if $\|x(t_0) - \bar{x}\| < \delta_1$, then $\|x(t) - \bar{x}\| < \epsilon, \forall t > t_0$.

2. **Attractivity.** $\exists \delta_2 > 0$ such that if $\|x(t_0) - \bar{x}\| < \delta_2$, then $x(t) \to \bar{x}$ as $t \to \infty$.

If only the first condition is satisfied, then the system is *globally stable*. If both conditions are satisfied only for some $\epsilon(x(0)) > 0$ then the stability properties hold only locally and the system is said to be *locally asymptotically stable*.

Local stability in a PI set $\mathcal{X}$ is equivalent to the existence of a $\mathcal{KL}$-function $\beta$ and a finite constant $\delta \geq 0$ such that:

$$\|x(k) - \bar{x}\| \leq \beta(\|x(0) - \bar{x}\|, k) + \delta, \ \forall x(0) \in \mathcal{X}, \ k \geq 0. \tag{9}$$

If $\delta = 0$ then the system is *asymptotically* stable. If the positively-invariant set is $\mathcal{X} = \mathbb{R}^n$ then stability holds *globally*.

Define the system output as $y(k) = \psi(x(k))$, where $\psi$ is a continuous, Lipschitz function. Input-to-Output stability provides a natural extension of asymptotic stability to systems with inputs or additive uncertainty[4].

**Definition A.8.** (Input-Output (practical) Stability) Given an RPI set $\mathcal{X}$, a constant nominal input $\bar{u}$ and a nominal steady state $\bar{x}(\bar{u}) \in \mathcal{X}$ such that $\bar{y} = \psi(\bar{x})$, the system Eq. (7) is said to be *input-output (practically) stable* to *bounded additive input perturbations* (IOpS) in $\mathcal{X}$ if there exists a $\mathcal{KL}$-function $\beta$ and a $\mathcal{K}_\infty$ function $\gamma$ and a constant $\zeta > 0$:

$$\|y(k) - \bar{y}\| \leq \beta(\|y(0) - \bar{y}\|, k) + \gamma(\|w\|) + \zeta, \ \forall x(0) \in \mathcal{X}, \tag{10}$$
$$u = \bar{u} + w, \ \bar{u} \in \mathcal{U}, \ \forall w \in \mathcal{W}, \ \forall k \geq 0.$$

**Definition A.9.** (Input-Output (Robust) Stability) Given an RPI set $\mathcal{X}$, a constant nominal input $\bar{u}$ and a nominal steady state $\bar{x}(\bar{u}) \in \mathcal{X}$ such that $\bar{y} = \psi(\bar{x})$, the system Eq. (7) is said to be *input-output (robustly) stable* to *bounded additive input perturbations* (IOS) in $\mathcal{X}$ if there exists a $\mathcal{KL}$-function $\beta$ and a $\mathcal{K}_\infty$ function $\gamma$ such that:

$$\|y(k) - \bar{y}\| \leq \beta(\|y(0) - \bar{y}\|, k) + \gamma(\|w\|), \ \forall x(0) \in \mathcal{X}, \tag{11}$$
$$u = \bar{u} + w, \ \bar{u} \in \mathcal{U}, \ \forall w \in \mathcal{W}, \ \forall k \geq 0.$$

## A.3 Jacobian Condition for Stability

The 1-step state transfer Jacobian for tied weights is:

$$J(x(k), u) = \frac{\partial f(x(k), u)}{\partial x(k)} = I + h\frac{\partial \sigma(\Delta x(k))}{\partial \Delta x(k)}A = I + h\sigma'(\Delta x(k))A. \qquad (12)$$

Recall, from the main paper, that the following stability condition is introduced:

**Condition 1.** (Condition 1 from *main paper*)

For any $\underline{\sigma} > 0$, the Jacobian satisfies:

$$\bar{\rho} = \sup_{(x,u)\in\mathcal{P}} \rho(J(x,u)) < 1, \qquad (13)$$

where $\rho(\cdot)$ is the spectral radius.

## A.4 Stability Result for Tied Weights

Stability of NAIS-Net is described in Theorem 1 from the main paper. For the sake of completeness, a longer version of this theorem is introduced here and will be proven in Section C. Note that proving the following result is equivalent to proving Theorem 1 in the main paper.

**Theorem 1.** (Asymptotic stability for shared weights)

If Condition 1 holds, then NAIS-Net with ReLU or $tanh$ activations is Asymptotically Stable with respect to *input dependent* equilibrium points. More formally:

$$x(k) \to \bar{x} \in \mathbb{R}^n, \ \forall x(0) \in \mathcal{X} \subseteq \mathbb{R}^n, \ u \in \mathbb{R}^m. \qquad (14)$$

The trajectory is described by:

$$\|x(k) - \bar{x}\| \le \bar{\rho}^k \|x(0) - \bar{x}\| \qquad (15)$$

where $\|\cdot\|$ is a suitable matrix norm and $\bar{\rho} < 1$ is given in Eq. (13).

In particular:

1. With $tanh$ activation, the steady state is independent of the initial state, namely, $x(k) \to \bar{x} \in \mathbb{R}^n, \ \forall x(0) \in \mathbb{R}^n$. The steady state is given by:

$$\bar{x} = -A^{-1}(Bu + b).$$

   The network is Globally Asymptotically Stable with respect to $\bar{x}$.

2. If the activation is $tanh$ then the network is Globally Input-Output (robustly) Stable for any input perturbation $w \in \mathbb{R}^m$. The trajectory is described by:

$$\|x(k) - \bar{x}\| \le \bar{\rho}^k \|x(0) - \bar{x}\| + \gamma(\|w\|), \qquad (16)$$

   with input-output gain

$$\gamma(\|w\|) = h\frac{\|B\|}{(1-\bar{\rho})}\|w\|. \qquad (17)$$

   If $\|w\| \le \mu$, then the following set is robustly positively invariant:

$$\mathcal{X} = \{x \in \mathbb{R}^n : \|x - \bar{x}\| \le \gamma(\mu)\}. \qquad (18)$$

   In other words:

$$x(0) \in \mathcal{X} \Rightarrow x(k) \in \mathcal{X}, \ \forall k > 0. \qquad (19)$$

3. In the case of ReLU activation, the network admits multiple equilibria, $\bar{x}$, and is Locally Asymptotically Stable with respect to each $\bar{x}$. In other words

$$x(k) \to \bar{x} \in \bar{\mathcal{X}} \subset \mathbb{R}^n, \ \forall x(0) \in \mathbb{R}^n.$$

   The steady state $\bar{x}$ is determined by a continuous piece-wise linear function of the input and the of initial condition. All network steady states are also contained in the set:

$$\bar{\mathcal{X}} = \{\bar{x} \in \mathbb{R}^n : A\bar{x} + Bu + b \le 0\}. \qquad (20)$$

   Moreover, if $A_{i\bullet}x + B_{i\bullet}u + b_i < 0$ is verified for some $i$ and for a finite unroll length, $\bar{k}_i \ge 0$, then we have that:

$$x_i(\bar{k}_i + t) = x_i(\bar{k}_i) = \bar{x}_i, \ \forall t \ge 0. \qquad (21)$$

4. If the activation is ReLU, then the network is Globally Input-Output practically Stable. The trajectory is described by:

$$\|x(k) - \bar{x}\| \le \bar{\rho}^k \|x(0) - \bar{x}\| + \gamma(\|w\|) + \zeta. \tag{22}$$

The input-output gain is given by Eq. (17), and the constant $\zeta > 0$ is the norm ball radius for the initial condition, namely,

$$\mathcal{X} = \{x \in \mathbb{R}^n : \|x - \bar{x}\| \le \zeta\}.$$

If the additive input perturbation satisfies $\|w\| \le \mu$, then the state converges to the ultimate bound:

$$\tilde{\mathcal{X}} = \{x \in \mathbb{R}^n : \|x - \bar{x}\| \le \zeta + \gamma(\mu)\}. \tag{23}$$

In other words:

$$\exists \tilde{\mathcal{X}}, \ \mathcal{X} \subset \tilde{\mathcal{X}} \subset \mathbb{R}^n : x(0) \in \mathcal{X} \Rightarrow x(k) \to \tilde{\mathcal{X}}, \tag{24}$$

for any input perturbations, $w$. The perturbed steady state for a constant $w$ depends also on the initial condition and it is a point in the set:

$$\tilde{\mathcal{X}}^w = \{\tilde{x} \in \mathbb{R}^n : A\tilde{x} + B(\bar{u} + w) + b \le 0\}, \tag{25}$$

where $u = \bar{u} + w$ and $\bar{u}$ is the nominal input.

## B    NAIS-Net with Untied Weights

### B.1    Proposed Network with Untied Weights

The proposed network architecture with skip connections and our robust stability results can be extended to the untied weight case. In particular, a single NAIS-Net block is analysed, where the weights are not shared throughout the unroll.

#### B.1.1    Fully Connected Layers

Consider in the following Deep ResNet with input skip connections and untied weights:

$$x(k+1) = f(x(k), u, k) = x(k) + h\sigma\Big(A(k)x(k) + B(k)u + b(k)\Big), \tag{26}$$

where $k$ indicates the layer, $u$ is the input data, $h > 0$, and $f$ is a continuous, differentiable function with bounded slope. The activation operator $\sigma$ is a vector of (element-wise) instances of a non-linear activation function. In the case of tied weights, the DNN Eq. (26) can be seen as a finite unroll of an RNN, where the layer index $k$ becomes a time index and the input is passed through the RNN at each time steps. This is fundamentally a linear difference equation also known as a discrete time dynamic system. The same can be said for the untied case with the difference that here the weights of the RNN will be time varying (this is a time varying dynamical system).

#### B.1.2    Convolutional Layers

For convolutional networks, the proposed layer architecture can be extended as:

$$\begin{aligned} X(k+1) &= F(X(k), U, k) \\ &= X(k) + h\sigma\Big(C(k) * (k) + D(k) * U + E(k)\Big), \end{aligned} \tag{27}$$

where $X^{(i)}(k)$, is the layer state matrix for channel $i$, while $U^{(j)}$, is the layer input data matrix for channel $j$ (where an appropriate zero padding has been applied) at layer $k$. An equivalent representation to Eq. (27), for a given layer $k$, can be computed in a similar way as done for the tied weight case in Appendix D.2. In particular, denote the matrix entries for the filter tensors $C(k)$ and $D(k)$ and $E(k)$ as follows: $C^{(c)}_{(i)}(k)$ as the state convolution filter from state channel $i$ to state channel $c$, and $D^{(c)}_{(j)}(k)$ is the input convolution filter from input channel $j$ to state channel $c$, and $E^{(c)}(k)$ is a bias matrix for the state channel $c$.

Once again, convolutional layers can be analysed in a similar way to fully connected layers, by means of the following vectorised representation:

$$x(k+1) = x(k) + h\sigma\big(A(k)x(k) + B(k)u + b(k)\big). \tag{28}$$

By means of the vectorised representation Eq. (28), the theoretical results proposed in this section will hold for both fully connected and convolutional layers.

## B.2 Non-autonomous set

Recall that, for a $\tanh$ activation, for each layer $k$ we have a different set $\mathcal{P}(k) \subseteq \mathbb{R}^{n+m}$ (with $\mathcal{P}(k) = \mathbb{R}^{n+m}$ for $\epsilon \to 0$). For ReLU activation, we have instead:

$$\mathcal{P} = \mathcal{P}(k) = \{(x,u) \in \mathbb{R}^n \times \mathbb{R}^m : A(k)x + B(k)u + b(k) > 0\}. \tag{29}$$

## B.3 Stability Definitions for Untied Weights

For the case of untied weights, let us consider the origin as a reference point ($\bar{x} = 0$, $\bar{u} = 0$) as no other steady state is possible without assuming convergence for $A(k)$, $B(k)$. This is true if $u = 0$ and if $b(k) = 0, \forall k \geq \bar{k} \geq 0$. The following definition is given for stability that is verified *uniformly* with respect to the changing weights:

**Definition B.1.** (Uniform Stability and Uniform Robustness) Consider $\bar{x} = 0$ and $\bar{u} = 0$. The network origin is said to be *uniformly* asymptotically or simply uniformly stable and, respectively, uniformly practically stable (IOpS) or uniformly Input-Output Stable (IOS) if, respectively, Definition A.7, A.8 and A.9 hold with a unique set of describing functions, $\beta$, $\gamma$, $\zeta$ for all possible values of the layer specific weights, $A(k)$, $B(k)$, $b(k)$, $\forall k \geq 0$.

## B.4 Jacobian Condition for Stability

The state transfer Jacobian for untied weights is:

$$J(x(k), u, k) = \frac{\partial f(x(k), u, k)}{\partial x(k)} = I + h\sigma' \left(\Delta x(k)\right) A(k). \tag{30}$$

The following assumption extends our results to the untied weight case:

**Condition 2.** For any $\bar{\sigma}' > 0$, the Jacobian satisfies:

$$\bar{\rho} = \sup_{(x,u)\in\mathcal{P}} \sup_k \rho\left(J(x(k), u, k)\right) < 1, \tag{31}$$

where $\rho(\cdot)$ is the spectral radius.

Condition Eq. (31) can be enforced during training for each layer using the procedures presented in the paper.

## B.5 Stability Result for Untied Weights

Recall that we have taken the origin as the reference equilibrium point, namely, $\bar{x} = 0$ is a steady state if $\bar{u} = 0$ and if $b(k) = 0, \forall k \geq \bar{k} \geq 0$. Without loss of generality, we will assume $b(k) = 0, \forall k$ and treat $u$ as a disturbance, $u = w$, for the robust case. The following result is obtained:

**Theorem 2.** (Main result for untied weights)

If Condition 2 holds, then NAIS–net with untied weights and with ReLU or $\tanh$ activations is Globally Uniformly Stable. In other words there is a set $\bar{\mathcal{X}}$ that is an ultimate bound, namely:

$$x(k) \to \bar{\mathcal{X}} \subseteq \mathbb{R}^n, \ \forall x(0) \in \mathcal{X} \subseteq \mathbb{R}^n. \tag{32}$$

The describing functions are:

$$\beta(\|x\|, k) = \bar{\rho}^k \|x\|, \qquad \gamma(\|w\|) = h\frac{\|\bar{B}\|}{(1-\bar{\rho})}\|w\|,$$
$$\bar{B} = \sup_k \|B(k)\|, \ \bar{\rho} < 1, \tag{33}$$

where $\|\cdot\|$ is the matrix norm providing the tightest bound to the left-hand side of Eq. (31), where $\bar{\rho}$ is defined.

In particular, we have:

1. If the activation is $\tanh$ then the network is Globally Uniformly Input-Output robustly Stable for any input perturbation $w \in \mathcal{W} = \mathbb{R}^m$. Under no input actions, namely if $u = 0$, then the origin is Globally Asymptotically Stable. If $u = w \in \mathcal{W}$ where $\mathcal{W}$ is the norm ball of radius $\mu$, then the following set is RPI:

$$\mathcal{X} = \left\{ x \in \mathbb{R}^n : \|x\| \leq \bar{r} = \frac{h\|\bar{B}\|}{1 - \bar{\rho}} \mu \right\}. \tag{34}$$

2. If the activation is ReLu then the network with zero nominal input is Globally Uniformly Input-Output practically Stable for input perturbations in a compact $\mathcal{W} \subset \mathbb{R}^m$. The describing function are given by Eq. (33) and the constant term is $\zeta = \bar{r}$, where $\bar{r}$ is the norm ball radius for the initial condition, namely,

$$\mathcal{X} = \{ x \in \mathbb{R}^n : \|x(0)\| \leq \bar{r} \}. \tag{35}$$

This set is PI under no input action.

If $u = w \in \mathcal{W}$ where $\mathcal{W}$ is the norm ball of radius $\mu$, then the state converges to the following ultimate bound:

$$\bar{\mathcal{X}} = \left\{ x \in \mathbb{R}^n : \|x\| \leq \bar{r} + h \frac{\|\bar{B}\|\mu}{(1 - \bar{\rho})} \right\}. \tag{36}$$

Note that, if the network consists of a combination of fully connected and convolutional layers, then a single norm inequality with corresponding $\beta$ and $\gamma$ can be obtained by means of matrix norm identities. For instance, since for any norm we have that $\|\cdot\|_q \leq \alpha \|\cdot\|_p$, with $\alpha > 0$, one could consider the global describing function $\beta(\cdot, \cdot) = \alpha \beta_p(\cdot, \cdot)$. Similarly for $\gamma$.

## C   Stability Proofs

Stability of a system can then be assessed for instance by use of the so-called Lyapunov indirect method [7], [4], [6]. Namely, if the linearised system around an equilibrium is stable then the original system is also stable. This also applies for linearisations around all possible trajectories if there is a single equilibrium point, as in our case. In the following, the bias term is sometimes omitted without loss of generality (one can add it to the input). Note that the proofs are also valid for RNNs with varying input sequences $u(k)$, with asymptotic results for converging input sequences, $u(k) \to \bar{u}$. Recall that $y(k) = x(k)$ by definition and let's consider the case of $b(k) = 0$, $\forall k$, without loss of generality as this can be also assumed as part of the input.

The untied case is covered first. The proposed results make use of the 1-step state transfer Jacobian Eq. (30). Note that this is not the same as the full input-to-output Jacobian, for instance as the one defined in [1]. The full Jacobian will also contain the input to state map, given by Eq. (39), which does not affect stability. The input to state map will be used later on to investigate robustness as well as the asymptotic network behaviour. First, we will focus on stability, which is determined by the 1-step state transfer map. For the sake of brevity, denote the layer $t$ state Jacobian from Eq. (30) as:

$$J(t) = I + h\sigma'(\Delta x(t))A(t), \ \forall t \geq 0.$$

Define the discrete time convolution sum as:

$$y_u(k) = \sum_{t=0}^{k-1} H(k - t)u(t), \ k > 0, \ y_u(0) = 0.$$

The above represent the *forced response* of a linear time invariant (LTI) system, namely the response to an input from a zero initial condition, where $H$ is the (in the LTI case stationary) impulse response. Conversely, the *free response* of an autonomous system from a non-zero initial condition is given by:

$$y_{x_0}(k) = \left( \prod_{t=0}^{k-1} J(t) \right) x(0).$$

The free response tends to zero for an asymptotically stable system. Considering linearised dynamics allows us to use superposition, in other words, the linearised system response can be analysed as the sum of the free and forced response:

$$y(k) = y_{x_0}(k) + y_u(k).$$

Note that this is not true for the original network, just for its linearisation.

For the considered network, the forced response of the linearised (time varying) system to an impulse at time $t$, evaluated at time $k$, is given by:

$$H(k, t) = h\sigma^{'}(\Delta x(t))B(t), \text{ if } k = t + 1, \tag{37}$$

$$H(k, t) = \left(\prod_{l=t}^{k-2} J(l+1)\right) h\sigma^{'}(\Delta x(t))B(t), \ \forall k \geq t + 2. \tag{38}$$

Therefore, the forced response of the linearised system is:

$$y_u(k) = h\sigma^{'}(x(k-1), u(k-1))B(k-1)u(k-1)$$
$$+ \sum_{t=0}^{k-2} \left(\prod_{l=t}^{k-2} J(l+1)\right) h\sigma^{'}(\Delta x(t))B(t)u(t)$$
$$= \sum_{t=0}^{k-1} H(k, t)u(t). \tag{39}$$

Note that:

$$\|H(k, t)\| \leq h \sup_{(x,u)\in\mathcal{P}, \ j} \|J(x, u, j)\|^{(k-t-1)} \|B(j)\|, \ \forall k, t \geq 0$$

since $\|\sigma^{'}\| \leq 1$.

To prove our results, we will use the fact that the network with $\tanh$ or ReLU activation is globally Lipschitz, and that the activation functions have Lipschitz constant $M = 1$ with respect to any norm. This follows from the fact that:

$$\sup_{\Delta x \in \mathbb{R}^n} \max_i |\sigma^{'}_{ii}(\Delta x)| = 1.$$

This means that the trajectory norm can be upper bounded inside $\mathcal{P} \subseteq \mathbb{R}^n \times \mathbb{R}^m$ by means of:

$$\|f(x, u, k)\| \leq \sup_{(x,u)\in\mathcal{P}} \sup_k \|J(x, u, k)\|\|x\| + \sup_{(x,u)\in\mathcal{P}} \sup_k \sum_{t=0}^{k-1} \|H(k, t)\|\|u\| \tag{40}$$

For a non-zero steady state $\bar{x}$ we can define the error function:

$$e(k) = x(k) - \bar{x}, \tag{41}$$

From the fact that the network equation is globally Lipschitz and that $f(\bar{x}, \bar{u}) = \bar{x}$, we can show that the function that maps $e(k)$ into $e(k+1)$ is also Lipschitz for all $k$. In particular, denoting $e(k+1)$ as simply $e^+$ and dropping the index $k$ for the sake of notation, we have the following:

$$\|e^+\| = \|f(x, \bar{u}) - \bar{x}\| = \|f(x, \bar{u}) - f(\bar{x}, \bar{u})\|$$
$$= \|x + h\sigma(Ax + B\bar{u} + b) - \bar{x} - h\sigma(A\bar{x} + B\bar{u} + b)\|$$
$$\leq \sup_{(x,u)\in\mathcal{P}} \sup_t \left(\|I + h\sigma^{'}(\Delta x(t))A\|\|x - \bar{x}\|\right) + h\|B\bar{u} - B\bar{u} + b - b\| \tag{42}$$
$$= \sup_{(x,u)\in\mathcal{P}} \sup_t \left(\|I + h\sigma^{'}(\Delta x(t))A\|\right) \|x - \bar{x}\|.$$

Note that, at the next step, we also have:

$$\|f(f(x,\bar{u}),\bar{u}) - \bar{x}\| = \|f(f(x,\bar{u}),\bar{u}) - f(f(\bar{x},\bar{u}),\bar{u})\|$$

$$\leq \sup_{(x,u)\in\mathcal{P}} \sup_t \left(\|I + h\sigma^{'}(\Delta x(t))A\|\right) \|f(x,\bar{u}) - f(\bar{x},\bar{u})\|$$

$$+ h\|B\bar{u} - B\bar{u} + b - b\|$$

$$= \sup_{(x,u)\in\mathcal{P}} \sup_t \left(\|I + h\sigma^{'}(\Delta x(t))A\|\right) \|f(x,\bar{u}) - f(\bar{x},\bar{u})\|$$

$$= \sup_{(x,u)\in\mathcal{P}} \sup_t \left(\|I + h\sigma^{'}(\Delta x(t))A\|\right) \|f(x,\bar{u}) - \bar{x}\|$$

$$\leq \sup_{(x,u)\in\mathcal{P}} \sup_t \left(\|I + h\sigma^{'}(\Delta x(t))A\|^2\right) \|x - \bar{x}\|. \tag{43}$$

and therefore, by induction it also follows that the trajectory at time $k$ satisfies:

$$\| \circ_{t=0}^k f(x,\bar{u}) - \bar{x}\| = \|f \circ f \circ \ldots f\left(f(x,\bar{u})\right) - \bar{x}\|$$

$$\leq \left(\sup_{(x,u)\in\mathcal{P}} \sup_t \|I + h\sigma^{'}(\Delta x(t))A\|\right)^k \|x - \bar{x}\| = \bar{\rho}^k\|x - \bar{x}\| \tag{44}$$

By definition of $\mathcal{P}$, from the above we have that, for $\tanh$ activation, the network trajectory with respect to an equilibrium point $\bar{x}$, can be upper bounded in norm by the trajectory of the linearization while in $\mathcal{P}$. In the limit case of $\bar{\sigma}^{'} \to 0$ the bound becomes *global* as $\mathcal{P} \to \mathbb{R}^n \times \mathbb{R}^m$. On the other hand, for ReLU activation the upper bounds are valid only *locally*, in $\mathcal{P}$ iself. These considerations will be used to prove our main results.

*Proof.* (Proof of Theorem 2: Main result for untied weights) In the untied weight case the network does not admit non-zero steady states, as the matrices $A(k)$ and $B(k)$ are not assumed to converge with increasing $k$. Let us therefore consider the origin as our reference point, namely, $(\bar{x} = 0, \bar{u} = 0)$. Therefore, for the robust results we will consider the input $u = w$. The proof can now proceed by norm bounding the superposition of the free and forced response. Recall that $y(k) = x(k)$ by definition and consider $b(k) = 0$, $\forall k$, without loss of generality. Now, if $(x,u) \in \mathcal{P}$, for the linearised system we have the following:

$$\|x(k) - \bar{x}\| = \|e(k)\|$$

$$\leq \sup_{(x,u)\in\mathcal{P}} \sup_j \|J(x,u,j)\|^k\|e(0)\| + \sum_{t=0}^{k-1} \|H(k,t)\| \|w(t)\|$$

$$\leq \bar{\rho}^k\|e(0)\| + h\sum_{t=0}^{k-1} \bar{\rho}^{k-t-1} \sup_j \|B(j)\|\|w\| \tag{45}$$

$$\leq \beta(\|e(0)\|,k) + \gamma(\|w\|).$$

In the above, we have defined:

$$\beta(\|e\|,k) = \bar{\rho}^k\|e\|,$$

$$\gamma(\|w\|) = h\frac{\|\bar{B}\|}{(1-\bar{\rho})}\|w\|, \tag{46}$$

$$\bar{B} = \sup_j \|B(j)\|, \ \bar{\rho} < 1.$$

In Eq. (45), we have used the fact that if $\rho(J) < 1$ then there exist a suitable matrix norm $\|\cdot\|$ and a constant $\bar{\rho} < 1$ such that $\|J\| \leq \bar{\rho}$. This stems directly from Theorem 5.6.12, page 298 of [3]. In our case, $\bar{\rho} < 1$ is verified when $(x,u) \in \mathcal{P}$ since, from Condition 2, we have that $\sup_{(x,u)\in\mathcal{P}} \sup_j \rho\left(J(x,u,j)\right) < 1$. Outside the region $\mathcal{P}$, however, we need to consider the specific activation functions: for $\tanh$, the region $\mathcal{P}$ can be taken to be any subset of the reals, therefore being outside this set as $\mathcal{P} \to \mathbb{R}^{n\times m}$ contradicts asymptotic stability. [5] For ReLU activation, being outside

$\mathcal{P}$ means that (at least part of) the system is autonomous and therefore the network is simply stable and Input-Output practically stable.

Theorem statements are proven as follows:

1. Global Stability follows from point 2 by taking $\bar{x} = 0$ and $\delta = \gamma(\|u\|)$.

2. Note that, the condition $(x(t), u) \notin \mathcal{P}$ is only possible for ReLU activations, since for $tanh$ activation we have that $\mathcal{P} \to \mathbb{R}^{n+m}$ for $\bar{\sigma}' \to 0$ and this would contradict stability result Eq. (45) inside the set. This means that Eq. (45) holds globally and therefore the considered network with tanh activations is Input-Output stable for a real-valued input $u$. Same considerations apply for the robust case with an additive perturbation $w \in \mathbb{R}^m$.

   In order to show the existence of a robust positively invariant set, consider the candidate set:

   $$\mathcal{X} = \left\{ x \in \mathbb{R}^n : \|e\| = \|x - \bar{x}\| \le \bar{r} = \frac{r}{1 - \bar{\rho}} \right\}, \quad (47)$$

   and the disturbance set:
   $$\mathcal{W} = \{ w \in \mathbb{R}^m : \|w\| \le \mu \}. \quad (48)$$

   Then from the IOS inequality we have that, if $x(0) \in \mathcal{X}$ and $w \in \mathcal{W}$, the bound $\mu$ can be computed so that $\mathcal{X}$ is RPI. To construct the bound, it is sufficient to have the following condition to hold $\forall k \ge 0$:

   $$\|e(k)\| \le \bar{\rho}^k \|e(0)\| + \frac{h\|B\|}{(1 - \bar{\rho})} \|w\| \le \bar{r}, \ \forall w : \|w\| \le \mu. \quad (49)$$

   The above is verified $\forall k \ge 0$ if the bound $\mu$ satisfies by the following sufficient condition when $\|e(0)\| \ge \bar{r}$:

   $$
   \begin{aligned}
   \|e(k)\| &\le \bar{\rho}^{k-1}\|e(0)\| + \bar{\rho}^{k-1}(\bar{\rho} - 1)\|e(0)\| + \frac{h\|\bar{B}\|}{(1 - \bar{\rho})}\|w\| \\
   &\le \bar{\rho}^{k-1}\|e(0)\| + (\bar{\rho} - 1)\|e(0)\| \sup_j \sum_{t=0}^{j} \bar{\rho}^{j-1} + \frac{h\|\bar{B}\|}{(1 - \bar{\rho})}\|w\| \\
   &= \bar{\rho}^{k-1}\|e(0)\| - \frac{(1 - \bar{\rho})}{(1 - \bar{\rho})}\|e(0)\| + \frac{h\|\bar{B}\|}{(1 - \bar{\rho})}\|w\| \\
   &\le \bar{\rho}^{k-1}\|e(0)\| \le \|e(0)\| \\
   &\le \bar{\rho}^{k-1}\bar{r} \le \bar{r}, \ \text{IF } \|e(0)\| = \bar{r} \\
   &\Leftarrow \frac{h\|\bar{B}\|}{(1 - \bar{\rho})}\|w\| \le \|e(0)\| \ge \bar{r} \\
   &\Leftarrow \frac{r}{(1 - \bar{\rho})} \ge \frac{h\|\bar{B}\|_2}{(1 - \bar{\rho})}\|u\| \\
   &\Leftrightarrow r \ge h\|\bar{B}\|\|w\| \\
   &\Leftarrow \|w\| \le \frac{r}{h\|\bar{B}\|} = \mu.
   \end{aligned}
   \quad (50)
   $$

   This will not necessarily hold in the interior of the set but it will hold outside and on the boundary of this compact set. Therefore the set $\mathcal{X}$ is invariant under $u = \bar{u} + w$, with $w \in \mathcal{W}$, namely, no solution starting inside $\mathcal{X}$ can pass its boundary under any $w \in \mathcal{W}$. Note that, for $tanh$ activations, this result holds globally. Conversely, given a bound $\mu$ for $\mathcal{W}$, we can compute $\bar{r}$ such that there is a set $\mathcal{X}$ that is RPI, namely:

   $$\mathcal{X} = \left\{ x \in \mathbb{R}^n : \|x\| \le \bar{r} = \frac{h\|\bar{B}\|}{1 - \bar{\rho}}\mu \right\}. \quad (51)$$

*every bounded subset of reals.* However, from the steady state analysis (to follow) and the contraction result we can see that $\sigma' = 0$ is highly unlikely to happen in practise.

3. Global Stability follows from point 4 by taking $\bar{x} = 0$ and $\delta = \gamma(\|u\|) + \zeta$.

4. For ReLU activations, the set $\mathcal{P}(k)$ does not cover the entire $\mathbb{R}^n$. This complicates the analysis further as the output $i$ of the network layer $k$ becomes autonomous when $(x(k), u) \notin \mathcal{P}_i(k)$. In particular, if at any $k = t$ we have $(x(t), u) \notin \mathcal{P}_i(t)$ then we also have that:

$$\begin{aligned} x_i(t+1) = x_i(t) \Rightarrow \|e(t+1)\| &\leq \beta(\|e(0)\|, t) + \gamma(\|w\|) \\ &\leq \beta(\|e(0)\|, t+1) + \gamma(\|w\|) + \zeta_t, \end{aligned} \tag{52}$$

where:

$$\zeta_t = \sup_{x(0) \in \mathcal{X}} \left( \beta(\|e(0)\|, t) - \beta(\|e(0)\|, t+1) \right) = \sup_{x(0) \in \mathcal{X}} \bar{\rho}^t (1 - \bar{\rho}) \|e(0)\|. \tag{53}$$

Where $\mathcal{X} = \{x \in \mathbb{R}^n : \|x - \bar{x}\| \leq \bar{r}\}$ is a bounded set of initial states. From the last two equations we can satisfy the IOpS condition, with:

$$\|e(k)\| \leq \beta(\|e(0)\|, k) + \gamma(\|w\|) + \zeta, \tag{54}$$

$$\zeta = \sup_k \sum_{j=0}^{k} \zeta_j = \frac{(1 - \bar{\rho})}{(1 - \bar{\rho})} \bar{r} = \bar{r}. \tag{55}$$

From the above we can notice that the set $\mathcal{X}$ is positively invariant if $u = \bar{u}$. If instead $u = \bar{u} + w$ with $w \in \mathcal{W} = \{w \in \mathbb{R}^m : \|w\| \leq \mu\}$ then, by taking the sup over this set of the gain $\gamma$, and over $k$ of the IOpS inequality Eq. (54), we have the ultimate bound $\bar{\mathcal{X}}$ defined by:

$$\bar{\mathcal{X}} = \left\{ x \in \mathbb{R}^n : \|x - \bar{x}\| \leq \bar{r} + h \frac{\|\bar{B}\|\mu}{(1 - \bar{\rho})} \right\}, \tag{56}$$

such that $x(0) \in \mathcal{X} \subset \bar{\mathcal{X}} \Rightarrow x(k) \to \bar{\mathcal{X}}$ as $k \to \infty$. Therefore, since $\bar{x} = 0$, for ReLU activations in the untied case we have global uniform Input-Output practical stability around the origin.

$\square$

We are now ready to prove Theorem 1 and its shorter version from the *main paper*.

*Proof.* (Proof of Theorem 1: Asymptotic stability for shared weights)

1. Recall that we have assumed that, for a $\tanh$ activation function, the network dynamics can be globally approximated by its linearisation. In the case of tied weights we have, for tanh activation, the steady state condition:

$$\bar{x} = \bar{x} + h \tanh(A\bar{x} + Bu + b) \Leftrightarrow A\bar{x} + Bu + b = 0. \tag{57}$$

The network has a unique input-dependant steady state $\bar{x}$ with steady state gain $G$ : $\|H(k)\| \to G$, given by, respectively[6]:

$$\bar{x} = -A^{-1}(Bu + b), \tag{58}$$

$$G = \left\| \frac{\partial \bar{x}}{\partial u} \right\| = \frac{\|B\|}{\|A\|}. \tag{59}$$

From this point and the above considerations in the proof of Theorem 2 we will now prove that in the case of tied weights, the network with $\tanh$ activation is Globally Asymptotically Stable with equilibrium $\bar{x}$ given by Eq. (58). In order to do this we will again use the linearized system and apply superposition of the free response with the forced response. In particular, from the proof of Theorem 2 part 1 and from Eq. (44) we that the free response of $x(k) - \bar{x}$, for any steady state $\bar{x}$, is norm bounded by:

$$\beta(\|e(k)\|) = \bar{\rho}^k \|x(0) - \bar{x}\|,$$

which vanishes asymptotically. Conversely, recall that the forced response (of the linearised system) to the input $u$ is given by the discrete convolution Eq. (39). This convolution converges for any chosen linearisation point as $\bar{\rho} < 1$. Therefore, by linearising around any $\bar{x}$, we have:

$$x(k) \to (I - J(\bar{x}, u))^{-1} h\sigma'(\bar{x}, u)(Bu + b) = -A^{-1}(Bu + b),$$

thus providing the desired Global Asymptotic Stability result.

2. From the fact that the network function is Lipschitz, the network is also Globally Input-Output Robustly Stable, as shown for the case of untied weights in the proof of Theorem 2, part 2. Moreover, from Eq. (44) we have that the IOS property holds also around any nominal equilibrium point $\bar{x}$.

3. For ReLU activations, the network dynamics is piece-wise linear, and sub-differentiable. In particular, the network has $2^n$ possible 1-step transitions, namely 2 possible ones for each dimension. We will proceed by enumeration of all possible dynamics transitions. First of all, we have that:

$$\mathcal{P} = \{(x, u) \in \mathbb{R}^n \times \mathbb{R}^m : Ax + Bu + b > 0\}, \tag{60}$$
$$\mathcal{P}_i = \{(x, u) \in \mathbb{R}^n \times \mathbb{R}^m : A_{i\bullet}x + B_{i\bullet}u + b_i > 0\}. \tag{61}$$

The network state at the next step is then given by:

$$x_i^+ = \begin{cases} x_i + h(A_{i\bullet}x + B_{i\bullet}u + b_i) & IF \ -A_{i\bullet}x \le (B_{i\bullet}u + b_i), \\ x_i & IF \ -A_{i\bullet}x > (B_{i\bullet}u + b_i). \end{cases} \tag{62}$$

The activation slope for the $i$-th coordinate is the set valued:

$$\sigma'_{ii}(\Delta x) \in \begin{cases} \{1\}, & IF \ -A_{i\bullet}x < (B_{i\bullet}u + b_i) \\ \{0\}, & IF \ -A_{i\bullet}x > (B_{i\bullet}u + b_i) \\ [0, 1], & IF \ -A_{i\bullet}x = (B_{i\bullet}u + b_i) \end{cases} \tag{63}$$

The Jacobian is again given by:

$$J(x, u) = I + \sigma'(\Delta x)A. \tag{64}$$

Clearly, if $(x(k), u) \in \mathcal{P}$, $\forall k$ then the system is linear and time invariant. In this region the system is linear and it has a unique, input-dependant steady state, given by:

$$\bar{x} = -A^{-1}(Bu + b), \tag{65}$$

which is a point on the the boundary of $\mathcal{P}$ for the limit case of $\bar{\sigma} = 0$.

In general, we cannot expect $(x(k), u) \in \mathcal{P}$, $\forall k$. We can instead expect that the network steady state will be either on the boundary of $\mathcal{P}$ or outside the set. In particular, if $(x(k), u) \notin \mathcal{P}$ for some $k$ then $\exists i : x_i(k + t) = x_i(k)$, $\forall t \ge 0$ and therefore part of the network states will become autonomous from layer $k$, making the network simply stable (not asymptotically stable). Consider the case in which, at some time $k$, we have $(x(k), u) \notin \mathcal{P}_i$ (for example in $\mathcal{N}_i$). In this case we have that $x_i(k + t) = x_i(k)$, $\forall t \ge 0$ and the network state will be free to change in its remaining dimensions with the $i$-th one being in steady state. For the remaining dimensions, we take out $x_i$ and consider its effect as an additional bias element. This will result in a smaller state space $\tilde{x} = [x_j, \ldots, x_n]^T$ except for $x_i$, with state transfer matrix $I + h\tilde{A}$, where $\tilde{A}$ consists of all elements of $A$, without the $i$-th row and column. Same for $\tilde{B}$ having the elements of $B$ except the $i$-th row. Note that $\tilde{A}$ still has eigenvalues in the same region as $A$ and is therefore negative definite and invertible. For this new system we have two possibilities: in the first case the trajectory stays in $\mathcal{N}_i$ all the time and the steady states for all dimensions except $i$ are given by:

$$\bar{\tilde{x}} = \tilde{A}^{-1}(\tilde{B}u + \tilde{b} + \bar{J}\bar{x}_i) \tag{66}$$

where $\bar{J}$ is a diagonal matrix containing the $i$-th column of $J$ except for the $i$-th element. The other possibility is that at some point we also have that $(x(k), u) \notin \mathcal{P}_j$ for some other $j$ (they also can be more than one at the time) in which case one can reduce the state space again and repeat as above. We have therefore shown that the system is Asymptotically Stable and

the trajectory tends to a point belonging to the set $\mathcal{X} = \{(x, u) \in \mathbb{R}^{n+m}, \; Ax + Bu + b \leq 0\}$. Note that the network trajectory depends on the initial condition, and the fact that $(x(k), u) \in \mathcal{P}_i$ at time $k$ is also dependent on $x(0)$. Therefore the steady-state will also depend on the initial condition[7], namely, $\bar{x}$ and the network is Locally Asymptotically stable with respect to the specific $\bar{x}$. Now, the set of $x(0)$ mapping into $\bar{x}$ given $u$ for which the system is either stable or autonomous is the full set of reals. Therefore the network is stable everywhere. Finally, since the network dynamics are continuous and piece-wise affine over a partition of adjacent polyhedra, the function determining the steady states is also continuous and piece-wise affine.

4. To prove Input-Output practical Stability, note that each combination of different vector fields that make up $f(x, u)$ provides a vector field that is Input-Output Stable. Moreover, $f$ is continuous and so is the map between $(x(0), u)$ and the steady state $\bar{x}$. We can therefore fix the steady state and take the worst case IOpS gain for each value of the vector field to provide suitable upper bounds for the IOpS definition to be satisfied[8] as in proof of Theorem 2, part 4.

$\square$

Note that the gain $\gamma(\cdot)$ can be used, for instance, as a regularizer to reduce the effect of input perturbations on the output of the network.

# D   Proof of Constraint Implementation

In this section, the proposed implementation for fully connected and convolutional layers is shown to be sufficient to fulfil the stability constraint on the Jacobian spectral radius.

## D.1   Proof of Fully Connected Implementation

Recall Algorithm 1 from the main paper. The following result is obtained:

**Lemma 2.** The Jacobian stability condition, $\rho(J(x, u)) < 1$, is satisfied $\forall (x, u) \in \mathcal{P}$ for the fully connected layer if $h \leq 1$ and Algorithm 1 is used.

Since Lemma 2 is equivalent to Theorem 2 from the main paper, the former will be proven next.

*Proof.* (Proof of Lemma 2)

Recall that Algorithm 1 results in the following operation being performed after each training epoch:

$$\tilde{R} \leftarrow \begin{cases} \sqrt{\delta} \dfrac{R}{\sqrt{\|R^T R\|_F}} & \text{IF } \|R^T R\|_F > \delta \\ R, & \text{Otherwise,} \end{cases} \tag{67}$$

with $\delta = 1 - 2\epsilon \in (0, 1)$. Recall that

$$A = -R^T R - \epsilon I. \tag{68}$$

Then, Eq. (67) is equivalent to the update:

$$\tilde{A} \leftarrow \begin{cases} -\tilde{R}^T \tilde{R} - \epsilon I = -(1 - 2\epsilon) \dfrac{R^T R}{\|R^T R\|_F} - \epsilon I, & \text{IF } \|R^T R\|_F > 1 - 2\epsilon \\ A, & \text{Otherwise} \end{cases} \tag{69}$$

Eq. (67) guarantees that $\|R^T R\|_F \leq (1 - 2\epsilon)$. From the fact that the Frobenius norm is an upper bound of the spectral norm and because of symmetry we have that:

$$\rho(R^T R) = \|R^T R\|_2 \leq \|R^T R\|_F \leq 1 - 2\epsilon. \tag{70}$$

Recall that, from Eq. (68), $A$ is negative definite and it only has real negative real eigenvalues. Recall also Lemma 1. Therefore, by applying the definition in Eq. (68) we have that $R = 0 \Rightarrow A = -\epsilon I$, then the eigenvalues of $hA$ are always located within the interval $[-h(1 - \epsilon), -h\epsilon]$. This means that:

$$\rho(hA) \leq h \max\{\epsilon, 1 - \epsilon\}. \tag{71}$$

To complete the proof, recall that the network Jacobian is:

$$J(x, u) = I + h\sigma'(\Delta x)A.$$

We will now look at the specific activation functions:

1. For ReLU activation, we simply have that $J(x, u) = I + hA$ in the set $\mathcal{P}$. Lemma 1 implies that $I + hA$ has only positive real eigenvalues located in $[1 - h(1 - \epsilon), 1 - h\epsilon]$ which, when $h \in (0, 1]$, implies that:

$$\bar{\rho} \leq \max\{1 - h(1 - \epsilon), 1 - h\epsilon\} < 1. \tag{72}$$

2. For $\tanh$ activations, since the matrix

$$\bar{A} = \frac{1}{2}\left(\sigma'(\Delta x)A + A^T\left(\sigma'(\Delta x)\right)^T\right) \tag{73}$$

is symmetric, $A$ is negative definite and $\sigma'(\cdot)$ is diagonal with entries $\sigma'_{ii}(\cdot) \in [\underline{\sigma}, 1]$ with $0 < \underline{\sigma} \ll 1$ when $(x, u) \in \mathcal{P}$, then $\bar{A}$ is also negative definite in this set. Therefore, in virtue of the observations at page 399-400 of [3], we have that the real part of the eigenvalues of $\sigma'(\Delta x)A$ is always less than zero in $\mathcal{P}$. Namely,

$$\text{RE}(\text{eig}(\sigma'(\Delta x)A)) < 0. \tag{74}$$

At the same time, by construction of $A$ and again thanks to $\sigma'_{ii}(\cdot) \in [\underline{\sigma}, 1]$ and $\sigma'_{ij}(\cdot) = 0$ if $i \neq j$, we have that:

$$\rho(\sigma'(\Delta x)A) \leq \|\sigma'(\Delta x)A\|_2 \leq \|\sigma'(\Delta x)\|_2\|A\|_2 \leq \|A\|_2 \leq 1 - \epsilon \tag{75}$$

From the above considerations the real part of the eigenvalues of $h\sigma'(\Delta x)A$ is in the interval $[-h(1 - \epsilon), -h\underline{\sigma}\epsilon]$. Assume $h \leq 1$.

Finally, we show that the Jacobian has only positive real eigenvalues. From Theorem 2.2 in [8], we know that $\sigma'(\Delta x)A$ is *similar* to:

$$\sigma'(\Delta x)^{-1/2}\sigma'(\Delta x)A\sigma'(\Delta x)^{1/2} = \sigma'(\Delta x)^{1/2}A\sigma'(\Delta x)^{1/2} \tag{76}$$

$$= \sigma'(\Delta x)^{1/2}A(\sigma'(\Delta x)^{1/2})^T \tag{77}$$

which is symmetric, negative definite and therefore has only negative real eigenvalues, provided that $(x, u) \in \mathcal{P}$. Since similarity implies eigenvalue equivalence, then $\sigma'(\Delta x)A$ has only negative real eigenvalues. Then, from Lemma 1, and from Eq. (74) and Eq. (75) we have that the eigenvalues of $J(x, u)$ are positive real and contained in $[1 - h(1 - \epsilon), 1 - h\underline{\sigma}\epsilon]$. Therefore we have that $\bar{\rho} \leq \max\{1 - h(1 - \epsilon), 1 - h\underline{\sigma}\epsilon\}$, which is less than 1 as $\underline{\sigma} > 0$ by definition.

$\square$

The less restrictive bound $\delta = 2(1 - \epsilon)$ with $h \leq 1$ is also sufficient for stability but it can result in trajectories that oscillate since it it does not constrain the eigenvalues to be positive real. Practically speaking the bound $\delta = 2(1 - \epsilon)$ has has proven sufficient for our MNIST experiments to be successful, however, we believe it is important to stress the difference between this and our proposed bound. In particular, our solution for fully connected layers leads to a critically damped system, i.e. to a monotonic trajectory for the 1D case. This means that we can expect the activations to behave monotonically both in time and in space. This behaviour is demonstrated in Section E. The additional regularity of the resulting function acts as a stronger regularisation on the network.

Note also that in the above proof we have shown that, in the case of ReLU, the 2-norm is suitable to prove stability. Moreover, if $h = 1$ and $\epsilon \leq 0.5$ then we have $\bar{\rho} = 1 - \epsilon$.

## D.2 Derivation of Convolutional Layer Implementation

### D.2.1 Mathematical derivation of the proposed algorithm

Denote the convolution operator in NAIS-Net, for each latent map $c$, as the following:

$$X^{(c)}(k+1) = X^{(c)}(k) + h\sigma\left(\Delta X^{(c)}(k)\right), \tag{78}$$

where:

$$\Delta X^{(c)}(k) = \sum_i C_{(i)}^{(c)} * X^{(i)}(k) + \sum_j D_{(j)}^{(c)} * U^{(j)} + E^{(c)}, \tag{79}$$

and where $X^{(i)}(k)$, is the layer state matrix for channel $i$, $U^{(j)}$, is the layer input data matrix for channel $j$ (where an appropriate zero padding has been applied) at layer $k$. The activation function, $\sigma$, is again applied element-wise. Recall also Algorithm 2 from the main paper. The following Lemma is obtained (equivalent to Theorem 3 from the main paper):

**Lemma 3.** If Algorithm 2 is used, then the Jacobian stability condition, $\rho(J(x,u)) < 1$, is satisfied $\forall (x,u) \in \mathcal{P}$ for the convolutional layer with $h \leq 1$.

The first step to obtain the result is to prove that the convolutional layer can be expressed as a suitable fully connected layer as in Lemma 1 from the main paper. This is shown next.

*Proof.* (Proof of Lemma 1 from the main paper)

For layer $k$, define $X(k)$ as a tall matrix containing all the state channel matrices $X^{(c)}(k)$ stacked vertically, namely:

$$X = \begin{pmatrix} X^{(1)} \\ X^{(2)} \\ \vdots \\ X^{(N_c)} \end{pmatrix}. \tag{80}$$

Similar considerations apply to the input matrix $U$ and the bias matrix $E$. Then, convolutional layers can be analysed in a similar way to fully connected layers, by means of the following vectorised representation:

$$x(k+1) = x(k) + h\sigma\left(Ax(k) + Bu + b\right), \tag{81}$$

where $x(k) \in \mathbb{R}^{n_X^2 \cdot N_c}$, $u = \mathbb{R}^{n_U^2 \cdot N_c}$, where $N_c$ is the number of channels, $n_X$ is the size of the latent space matrix for a single channel, while $n_U$ is the size of the input data matrix for a single channel. In eq. (81), the matrices $A$ and $B$ are made of blocks containing the convolution filters elements in a particular structure, to be characterised next. First, in order to preserve dimensionality of $x$, the convolution for $x$ will have a fixed stride of 1, a filter size $n_C$ and a zero padding of $p \in \mathbb{N}$, such that $n_C = 2p + 1$. If a greater stride is used, then the state space can be extended with an appropriate number of constant zero entries (not connected). Let's then consider, without loss of generality, a unitary stride. The matrix $A$ is all we need to define in order to prove the Lemma.

In eq. (81), the vector $x$ is chosen to be the vectorised version of $X$. In particular,

$$x = \begin{pmatrix} x^{(1)} \\ x^{(2)} \\ \vdots \\ x^{(N_c)} \end{pmatrix}, \tag{82}$$

where $x^{(c)}$ is the vectorised version of $X^{(c)}$. More specifically, these objects are defined as:

$$X^{(c)} = \begin{pmatrix} X_{1,1}^{(c)} & X_{1,2}^{(c)} & \cdots & X_{1,n}^{(c)} \\ X_{2,1}^{(c)} & X_{2,2}^{(c)} & \cdots & X_{2,n}^{(c)} \\ \vdots & \vdots & \ddots & \vdots \\ X_{n,1}^{(c)} & X_{n,2}^{(c)} & \cdots & X_{n,n}^{(c)} \end{pmatrix} \in \mathbb{R}^{n_X \times n_X}, \tag{83}$$

and

$$x^{(c)} = \begin{pmatrix} X_{1,1}^{(c)} \\ X_{1,2}^{(c)} \\ \vdots \\ X_{n,n}^{(c)} \end{pmatrix} \in \mathbb{R}^{n_X^2}. \tag{84}$$

Similar considerations apply to $u$. The matrix $A$ in Eq. (81) has the following structure:

$$A = \begin{pmatrix} A_{(1)}^{(1)} & A_{(2)}^{(1)} & \cdots & A_{(Nc)}^{(1)} \\ A_{(1)}^{(2)} & A_{(2)}^{(2)} & \cdots & A_{(Nc)}^{(2)} \\ \vdots & \vdots & \ddots & \vdots \\ A_{(1)}^{(Nc)} & A_{(2)}^{(Nc)} & \cdots & A_{(Nc)}^{(Nc)} \end{pmatrix} \in \mathbb{R}^{(n^2 \cdot N_c) \times (n^2 \cdot N_c)}, \tag{85}$$

where $Nc$ is the number of channels and $A_{(i)}^{(c)}$ corresponds to the filter $C_{(i)}^{(c)}$. In particular, each row of $A$ contains in fact the elements of the filter $C_{(i)}^{(c)}$, plus some zero elements, with the central element of the filter $C_{(i)}^{(c)}$ on the diagonal. The latter point is instrumental to the proof and can be demonstrated as follows. Define the single channel filters as:

$$C_{(i)}^{(c)} = \begin{pmatrix} C_{(i)\,1,1}^{(c)} & C_{(i)\,1,2}^{(c)} & \cdots & C_{(i)\,1,n_C}^{(c)} \\ C_{(i)\,2,1}^{(c)} & C_{(i)\,2,2}^{(c)} & \cdots & C_{(i)\,2,n_C}^{(c)} \\ \vdots & \vdots & \ddots & \vdots \\ C_{(i)\,n_C,1}^{(c)} & C_{(i)\,n_C,2}^{(c)} & \cdots & C_{(i)\,n_C,n_C}^{(c)} \end{pmatrix}. \tag{86}$$

Consider now the output of the single channel convolution $Z_{(i)}^{(c)} = C_{(i)}^{(c)} * X^{(i)}$ with the discussed padding and stride. The first element of the resulting matrix, $Z_{(i)\,1,1}^{(c)}$ is determined by applying the filter to the first patch of $X^{(i)}$, suitably padded. For instance, for a 3-by-3 filter ($p = 1$), we have:

$$\text{patch}_{1,1}\left(X^{(i)}\right) = \begin{pmatrix} \boxed{\begin{matrix} 0 & 0 & 0 \\ 0 & X_{1,1}^{(c)} & X_{1,2}^{(c)} \\ 0 & X_{2,1}^{(c)} & X_{2,2}^{(c)} \end{matrix}} & \begin{matrix} 0 & \cdots & 0 \\ X_{1,n}^{(c)} & & 0 \\ X_{2,n}^{(c)} & & 0 \end{matrix} \\ \begin{matrix} \vdots & \vdots & \vdots \\ 0 & X_{n,1}^{(c)} & X_{n,2}^{(c)} \end{matrix} & \begin{matrix} \ddots & \vdots & \vdots \\ \cdots & X_{n,n}^{(c)} & 0 \end{matrix} \end{pmatrix}. \tag{87}$$

The first element of $Z_{(i)}^{(c)}$ is therefore given by:

$$Z_{(i)\,1,1}^{(c)} = X_{1,1}^{(i)} C_{(i)\,i_{\text{centre}},i_{\text{centre}}}^{(c)} + X_{1,2}^{(i)} C_{(i)\,i_{\text{centre}},i_{\text{centre}}+1}^{(c)} + \cdots + X_{n_C-p,n_C-p}^{(i)} C_{(i)\,n_C,n_C}^{(c)}, \tag{88}$$

where $i_{\text{centre}}$ denotes the central row (and column) of the filter.

The second element of the first row can be computed by means of the following patch (again when $p = 1$, for illustration):

$$\text{patch}_{1,2}\left(X^{(i)}\right) = \begin{pmatrix} \begin{matrix} 0 \\ 0 \\ 0 \end{matrix} & \boxed{\begin{matrix} 0 & 0 & 0 \\ X_{1,1}^{(c)} & X_{1,2}^{(c)} & \cdots \\ X_{2,1}^{(c)} & X_{2,2}^{(c)} & \cdots \end{matrix}} & \begin{matrix} \cdots & 0 \\ X_{1,n}^{(c)} & 0 \\ X_{2,n}^{(c)} & 0 \end{matrix} \\ \begin{matrix} \vdots \\ 0 \end{matrix} & \begin{matrix} \vdots & \vdots \\ X_{n,1}^{(c)} & X_{n,2}^{(c)} \end{matrix} & \begin{matrix} \ddots & \vdots & \vdots \\ \cdots & X_{n,n}^{(c)} & 0 \end{matrix} \end{pmatrix}. \tag{89}$$

The element is therefore given by:

$$Z_{(i)\,1,2}^{(c)} = X_{1,1}^{(i)} C_{(i)\,i_{\text{centre}},i_{\text{centre}}-1}^{(c)} + X_{1,2}^{(i)} C_{(i)\,i_{\text{centre}},i_{\text{centre}}}^{(c)} + \cdots + X_{n_C-p,n_C}^{(i)} C_{(i)\,n_C,n_C}^{(c)}. \tag{90}$$

The remaining elements of $Z_{(i)}^{(c)}$ will follow a similar rule, which will involve (in the worst case) all of the elements of the filter. In particular, one can notice for $Z_{ij}$ the corresponding element of $X$, $X_{ij}$, is always multiplied by $C_{(i)}^{(c)}{}_{i_{\text{centre}}, i_{\text{centre}}}$. In order to produce the matrix $A$, we can consider the Jacobian of $Z$. In particular, for the first two rows of $A_{(i)}^{(c)}$ we have:

$$A_{(i)\,1,\bullet}^{(c)} = \frac{\partial Z_{(i)\,1,1}^{(c)}}{\partial X^{(i)}} = \left[\ C_{(i)\,i_{\text{centre}}, i_{\text{centre}}}^{(c)} \quad C_{(i)\,i_{\text{centre}}+1, i_{\text{centre}}+1}^{(c)} \quad \cdots \quad C_{(i)\,n_C, n_C}^{(c)} \quad 0\ \right], \quad (91)$$

where $[\cdot]$ is used to define a row vector, and where $A_{(i)\,1,\bullet}^{(c)}$ has an appropriate number of zeros at the end, and

$$A_{(i)\,2,\bullet}^{(c)} = \frac{\partial Z_{(i)\,1,2}^{(c)}}{\partial X^{(i)}} = \left[\ C_{(i)\,i_{\text{centre}}, i_{\text{centre}}-1}^{(c)} \quad C_{(i)\,i_{\text{centre}}, i_{\text{centre}}}^{(c)} \quad \cdots \quad C_{(i)\,n_C, n_C}^{(c)} \quad 0\ \right]. \quad (92)$$

Note that, the vectors defined in eq. (91) and eq. (92), contain several zeros among the non-zero elements. By applying the filter $C_{(i)}^{(c)}$ to the remaining patches of $X^{(i)}$ one can inductively construct the matrix $A_{(i)}^{(c)}$. It can also be can noticed that each row $A_{(i)\,j,\bullet}^{(c)}$ contains *at most* all of the elements of the filter, with the central element of the filter in position $j$. By stacking together the obtained rows $A_{(i)\,j,\bullet}^{(c)}$ we obtain a matrix, $A_{(i)}^{(c)}$, which has $C_{(i)\,i_{\text{centre}}, i_{\text{centre}}}^{(c)}$ on the diagonal. Each row of this matrix contains, in the worst case, all of the elements of $C_{(i)}^{(c)}$.

Define the vectorised version of $Z_{(i)}^{(c)}$ as:

$$z_{(i)}^{(c)} = \begin{pmatrix} Z_{(i)\,1,1}^{(c)} \\ Z_{(i)\,1,2}^{(c)} \\ \vdots \\ Z_{(i)\,n,n}^{(c)} \end{pmatrix} \in \mathbb{R}^{n^2}, \quad (93)$$

which, by linearity, satisfies:

$$z_{(i)}^{(c)} = A_{(i)}^{(c)} x^{(i)}. \quad (94)$$

By summing over the index $i$ we obtain the vectorised output of the convolution $C * X$ for the channel $c$:

$$z^{(c)} = \sum_{i=1}^{N_c} A_{(i)}^{(c)} x^{(i)} = \left[\ A_{(1)}^{(c)} \quad A_{(2)}^{(c)} \quad \cdots \quad A_{(N_c)}^{(c)}\ \right] x = A^{(c)} x, \quad (95)$$

where the matrices $A_{(i)}^{(c)}$ are stacked horizontally and where we have used the definition of $x$ given in eq. (82). The full matrix $A$ is therefore given by:

$$A = \begin{pmatrix} A^{(1)} \\ A^{(2)} \\ \vdots \\ A^{(N_c)} \end{pmatrix} = \left( \begin{array}{cccc} A_{(1)}^{(1)} & A_{(2)}^{(1)} & \cdots & A_{(N_c)}^{(1)} \\ \hline A_{(1)}^{(2)} & A_{(2)}^{(2)} & \cdots & A_{(N_c)}^{(2)} \\ \hline \vdots & \vdots & \ddots & \vdots \\ \hline A_{(1)}^{(N_c)} & A_{(2)}^{(N_c)} & \cdots & A_{(N_c)}^{(N_c)} \end{array} \right), \quad (96)$$

where we have conveniently separated the long blocks used to produce the single channels result, $z^{(c)}$. By defining the vector

$$z = \begin{pmatrix} z^{(1)} \\ z^{(2)} \\ \vdots \\ z^{(N_c)} \end{pmatrix}, \quad (97)$$

and by means of eq. (95) we have that $z = Ax$. This is a vectorised representation of $C * X$. $\qquad \square$

We are now ready to prove Lemma 3 and consequently Theorem 3 from the main paper.

*Proof.* (Proof of Lemma 3)

Similar to what done for the fully connected layer, we will first show that the Algorithm 2 places the eigenvalues of $I + A$ strictly inside the unit circle. Then, we will also show that $J(x, u)$ enjoys the same property in $\mathcal{P}$.

We will now derive the steps used in Algorithm 2 to enforce that $\rho(I + A) \leq 1 - \epsilon$. Recall that, from Lemma 1, we need the eigenvalues $A$ to be lying inside the circle, $\mathcal{S}$, of the complex plane centered at $(-1, 0\jmath)$ with radius $1 - \epsilon$. More formally:

$$\mathcal{S} = \{\lambda \in \mathbb{C} : |\lambda + 1| \leq 1 - \epsilon\}. \tag{98}$$

The Gershgorin theorem offers a way to locate the eigenvalues of $A$ inside $\mathcal{S}$. Consider the $c$-th long long block in eq. (96), $A^{(c)}$. Recall the particular structure of $A^{(c)}_{(c)}$ as highlighted in eq. (91) and eq. (92). For each long block $A^{(c)}$ we have that all associated Gershgorin disks must satisfy:

$$\left| \lambda - C^{(c)}_{i_{\text{centre}}} \right| \leq \sum_{i \neq i_{\text{centre}}} \left| C^{(c)}_i \right|, \tag{99}$$

where $\lambda$ is any of the corresponding eigenvalues, $C^{(c)}_{i_{\text{centre}}}$ is the *central element* of the filter $C^{(c)}_{(c)}$, namely, $C^{(c)}_{i_{\text{centre}}} = C^{(c)}_{(c)}{}_{i_{\text{centre}}, i_{\text{centre}}}$, and the sum is performed over all of the remaining elements of $C^{(c)}_{(c)}$ plus all the elements of the remaining filters used for, $z^{(c)}$, namely $C^{(c)}_{(j)}, \forall j \neq c$. Therefore, one can act directly on the kernel $C$ without computing $A$.

By means of Algorithm 2, for each channel $c$ we have that:

$$C^{(c)}_{i_{\text{centre}}} = -1 - \delta_c, \tag{100}$$

and

$$\sum_{i \neq i_{\text{centre}}} \left| C^{(c)}_i \right| \leq 1 - \epsilon - |\delta_c| \tag{101}$$

where

$$|\delta_c| < 1 - \eta, \ 0 < \epsilon < \eta < 1. \tag{102}$$

This means that for every $c$, the $c$-th block of $A$ has the largest Gershgorin region bounded by a disk, $\mathcal{S}^c$. This disk is centred at $(-1 - \delta_c, 0\jmath)$ with radius $1 - \epsilon - |\delta_c|$. More formally, the disk is defined as:

$$\mathcal{S}^c = \{\lambda \in \mathbb{C} : |\lambda + 1 + \delta_c| \leq 1 - \epsilon - |\delta_c|\}. \tag{103}$$

Thanks to eq. (102), this region is not empty and its center can be only strictly inside $\mathcal{S}$. Clearly, we have that, $\mathcal{S}^c \subseteq \mathcal{S}, \forall c$. This follows from convexity and from the fact that, when $|\delta_c| = 1 - \eta$, thanks to eq. (102) we have

$$\mathcal{S}^c = \{\lambda \in \mathbb{C} : |\lambda + 1 \pm (1 - \eta)| \leq 1 - \epsilon - (1 - \eta)\} \subset \mathcal{S} \tag{104}$$

while on the other hand $\delta_c = 0 \Rightarrow \mathcal{S}^c = \mathcal{S}$.

We will now show that the eigenvalue condition also applies to the network state Jacobian. From Algorithm 2 we have that each row $j$ of the matrix $\underline{J} = I + A$ satisfies (for the corresponding channel $c$):

$$\sum_i |\underline{J}_{ji}| \leq \left| 1 + C^{(c)}_{i_{\text{centre}}} \right| + \sum_{i \neq i_{\text{centre}}} \left| C^{(c)}_i \right|$$
$$\leq |\delta_c| + 1 - \epsilon - |\delta_c| = 1 - \epsilon. \tag{105}$$

Thus $\|\underline{J}\|_\infty = \|I + A\|_\infty \leq 1 - \epsilon < 1$. [9] Recall that, in the vectorised representation, the state Jacobian is:

$$J(x, u) = I + h\sigma'(\Delta x)A.$$

Then, if $h \leq 1$, we have that $\forall (x, u) \in \mathcal{P}$:

$$\|J(x, u)\|_\infty = \max_i |1 - h\sigma'_{ii}(\Delta x)(1 + \delta_c)| + h\sigma'_{ii}(\Delta x) \sum_{j \neq i} |A_{ij}|$$

$$\leq \max_i \{1 - h\sigma'_{ii}(\Delta x) + h\sigma'_{ii}(\Delta x)|\delta_c| + h\sigma'_{ii}(\Delta x)(1 - \epsilon) - h\sigma'_{ii}(\Delta x)|\delta_c|\} \quad (106)$$

$$= \max_i \{1 - h\sigma'_{ii}(\Delta x)\epsilon\} < 1 - h\underline{\sigma}\epsilon \leq 1.$$

The proof is concluded by means of the identity $\rho(\cdot) \leq \|\cdot\|$. $\qquad\square$

Note that in the above we have also showed that in the convolutional case the infinity norm is suitable to prove stability. Moreover, for ReLU we have that $\bar{\rho} \leq 1 - h\epsilon$.

The above results can directly be extended to the untied weight case providing that the projections are applied at each stage of the unroll.

In the main paper we have used $\delta_c = 0$, $\forall c$ (equivalently, $\eta = 1$). This means that one filter weight per latent channel is fixed at $-1$ which might seem conservative. It is however worth noting that this choice provides the biggest Gershgorin disk for the matrix $A$ as defined in Eq. (104). Hence $\delta_c = 0$, $\forall c$ results in the least restriction for the remaining elements of the filter bank. At the same time, we have $N_C$ less parameters to train. A tradeoff is however possible if one wants to experiment with different values of $\eta \in (\epsilon, 1)$.

### D.2.2 Illustrative Example

Consider, for instance, 3 latent channels $X^{(1)}, \ldots, X^{(3)}$, which results in 3 blocks of 3 filters, 1 block per channel $X^{(c)}(k + 1)$. Each filter has the same size $n_C$, which again needs to be chosen such that the channel dimensionality is preserved. This corresponds to the following matrix:

$$
A = \left(
\begin{array}{c|c|c}
A^{(1)}_{(1)} & A^{(1)}_{(2)} & A^{(1)}_{(3)} \\ \hline
A^{(2)}_{(1)} & A^{(2)}_{(2)} & A^{(2)}_{(3)} \\ \hline
A^{(3)}_{(1)} & A^{(3)}_{(2)} & A^{(3)}_{(3)}
\end{array}
\right) =
$$

$$
= \left(
\begin{array}{cc|ccc|cc}
C^{(1)}_{i_{\text{centre}}} & C^{(1)}_j & C^{(1)}_{n_C-1} & \cdots & & \cdots & C^{(1)}_{3 \cdot n_C - 1} \\
\vdots & \vdots & \ddots & \cdots & & \cdots & \vdots \\ \hline
\tilde{C}^{(2)}_1 & \cdots & C^{(2)}_{i_{\text{centre}}} & \cdots & & \cdots & C^{(2)}_{3 \cdot c_C - 1} \\
\vdots & \vdots & \ddots & \cdots & & \cdots & \vdots \\ \hline
\tilde{C}^{(3)}_1 & \cdots & \cdots & \cdots & & C^{(3)}_{i_{\text{centre}}} & \vdots \\
\vdots & \vdots & \ddots & \cdots & & \cdots & \ddots
\end{array}
\right), \quad (107)
$$

where the relevant rows include all elements of the filters (in the worst case) together with a large number of zeros, the position of which does not matter for our results, and the diagonal elements are known and non-zero. Therefore, according to eq. (107), we have to repeat Algorithm 2 from *main paper* for each of the 3 filter banks.

# E  Analysis of 1D Activations

This paper has discussed the formulation and implementation of stability conditions for a non-autonomous dynamical system, inspired by the ResNet, with the addition of a necessary input skip connection. This system's unroll, given a constant input, results in a novel architecture for supervised learning. In particular, we are interested in using the system's trajectory in two ways. The first consists in using the last point of an unroll of fixed length $K$. The second one uses an unroll of varying length less or equal to $K > 0$, which depends on a stopping criterion $\|\Delta x(x, u)\| < \epsilon$, with $\epsilon > 0$. In the former case, this produces an input-output map (a NAIS-Net block) that is Lipschitz for any $K$ (even infinity) and significantly better behaved than a standard ResNet without the use regularisation or batch normalisation. The latter case, defines a *pattern-dependent processing depth* where the network results instead in a piecewise Lipschitz function for any $K > 0$ and $\epsilon > 0$.

In order to clarify further the role of stability, the behaviour of a scalar NAIS-Net block and its unstable version, *non-autonomous ResNet*, are compared. Recall that the NAIS-Net block is defined by the unroll of the dynamical system:

$$x(k+1) = x(k) + \Delta x(x(k), u(k)) = x(k) + h\sigma\Big(Ax(k) + Bu + b\Big). \qquad (108)$$

In this Section, since the scalar case is considered, $A$, $B$ and $b$ are scalar. In particular, we investigate different values of $A$ that satisfy or violate our stability conditions, namely $\|I + A\| < 1$, which in this special case translate into $A \in (-2, 0)$. Whenever this condition is violated we denote the resulting network as *non-autonomous ResNet* or, equivalently, *unstable NAIS-Net*. The input parameter $B$ is set to $1$ and the bias $b$ is set to zero for all experiments. Note that, in this setting, varying the input is equivalent to varying the bias.

## E.1  Fixed number of unroll steps

Figure 1 shows a couple of pathological cases in which the unstable non-autonomous ResNet produces unreliable or uninformative input-output maps. In particular, the top graphs present the case of positive unstable eigenvalues. The top left figure shows that in this case the tanh activated network presents bifurcations which can make gradients explode [5]. The slope is nearly infinite around the origin and finally, for large inputs, the activation collapses into a flat function equal to $kA$, where $k$ is the iteration number. The ReLU network (top right) has an exponential gain increase per step $k$ and the gain for $k = 100$ reaches $10^{30}$ (see red box and pointer). Bottom graphs present the case of large negative eigenvalues. In the bottom left figure, the tanh activation produces a map that has limited slope but is also quite irregular, especially around the origin. This can also result into large gradients during training. The ReLU activated network (bottom right) instead produces an uninformative map, $\max(u, 0)$, which is (locally) independent from the parameter $A$ and the unroll length $K$.

Figure 2 shows the input-output maps produced by stable NAIS-Net with our proposed reprojection for fully connected architectures. In particular, the top and bottom graphs present the case of positive real stable eigenvalues with different magnitude. This means that the resulting trajectories are critically damped, namely, oscillation-free. The left figures shows that as a result the stable tanh activated networks have monotonic activations (strictly increasing around the origin) that tend to a straight line as $k \to \infty$. This is confirmed by the theoretical results. Moreover, the map is Lipschitz with Lipschitz constant equal to the steady state gain presented in the main paper, $\|A^{-1}\| \, \|B\|$. Figures on the right present the ReLU case where the maps remain of the same form but change in slope until the theoretical gain $\|A^{-1}\| \, \|B\|$ is reached. This means that one cannot have an unbounded slope or the same map for different unroll length. The rate of change of the map as a function of $k$ is also determined by the parameters but it is always under control. Although this was not proven in our results, the behaviour of the resulting input-output maps suggests that gradient should be well behaved during training and not explode nor vanish. Finally, one could make the Lipschitz constant unitary by multiplying the network output by $\|A\|/\|B\|$ once the recursion is finished. This could be used as an alternative to batch normalization and it will be investigated in future.

Figure 3 shows the maps produced by stable NAIS-Net with eigenvalues that are outside the region of our proposed reprojection for fully connected layers but still inside the one for convolutional layers. In particular, the top and bottom graphs present the case of negative real stable eigenvalues with different magnitude. The left figures shows that as a result the stable tanh activated networks

Figure 1: **Single neuron unstable NAIS-Net (non-autonomous ResNet). Input-output map for different unroll length for tanh (Left) and ReLU (Right) activations.** Pathological cases in which the unstable non-autonomous ResNet produces unreliable or uninformative input-output maps. In particular, the top graphs present the case of positive unstable eigenvalues. The top left figure shows that in this case the tanh activated network presents bifurcations which can make gradients explode [5]. The slope is also nearly infinite around the origin. Elsewhere, the activation collapses into a flat function equal to $kA$, where $k$ is the iteration number. The ReLU activations (top right) has an exponential gain increase per step $k$ and the gain for $k = 100$ reaches $10^{30}$ (see red box and pointer). Bottom graphs present the case of large negative eigenvalues. In the bottom left figure, the tanh activation produces a map that has limited slope but it is quite irregular, especially around the origin. This can also result large gradients during training. The ReLU activated network (bottom right) instead produces an uninformative map, $\max(u, 0)$, locally independent from $A$ and the unroll length $K$.

have monotonic activations (not strictly in this case) that still tend to a straight line as expected but present intermediate decaying oscillations as $k \to \infty$. The map is still Lipschitz but the Lipschitz constant is not always equal to the steady state gain because of the transient oscillations. Figures on the right present the ReLU case where the maps remain identical independently of the parameter $A$. This means that this parameter becomes uninformative and one could argue that in this case gradients would vanish. Note that our reprojection for convolutional layers can theoretically allow for this behaviour while the fully connected version does not. Training with Algorithm 2 has been quite successful on our experiments, however, the above points are of interest and will be further investigated in follow-up work.

## E.2 Pattern-dependent processing depth through a stopping criterion

We investigate the use at test time of a stopping criterion for the network unroll, $\|\Delta x(x, u)\| \le \epsilon$, where $\epsilon$ is a hyper-parameter. The activations for a 1D network are again considered, where $\epsilon$ is set to $0.95$ for illustrative purpose. The resulting activations are discontinuous but locally preserve the properties illustrated in the previous Section.

Figure 4 shows the input-output maps produced by stable NAIS-Net with our proposed reprojection for fully connected architectures. In particular, the top and bottom graphs present the case of positive

Figure 2: **Single neuron NAIS-Net. Input-output map for different unroll length for tanh (Left) and ReLU (Right) activations.** The input-output maps produced by stable NAIS-Net with our proposed reprojection for fully connected architectures. In particular, the top and bottom graphs present the case of positive stable eigenvalues with different magnitude. The left figures shows that the stable tanh activated networks have monotonic activations (strictly increasing around the origin) that tend to a straight line as $k \to \infty$ as in our theoretical results. Morever, the map is Lipschitz with Lipschitz constant equal to the steady state gain presented in the main paper, $\|A^{-1}\| \|B\|$. Figures on the right present the ReLU case where the maps remain of the changes in slope up to $\|A^{-1}\| \|B\|$. This means that one cannot have an unbounded slope or the same map for different unroll length. The rate of change of the map changes as a function of $k$ is also determined by the parameters.

stable eigenvalues with different magnitude. The left figures shows that the stable tanh activated networks have piece-wise continuous and locally strictly monotonic activations (strictly increasing around the origin) that tend to a straight line as $k \to \infty$ as in our theoretical results. Moreover, the map is also piece-wise Lipschitz with Lipschitz constant less than the steady state gain presented in the main paper, $\|A^{-1}\| \|B\|$. Figures on the right present the ReLU case where the maps are piece-wise linear functions with slope that is upper bounded by $\|A^{-1}\| \|B\|$. This means that one cannot have an unbounded slope (except for the jumps) or the same map for different unroll length. The rate of change of the map changes as a function of $k$ is also determined by the parameters. The jump magnitude and the slopes are also dependant on the choice of threshold for the stopping criteria.

Figure 3: **Single neuron NAIS-Net with less conservative reprojection. Input-output map for different unroll length for tanh (Left) and ReLU (Right) activations.** Input-output maps produced by stable NAIS-Net with eigenvalues that are outside the region of our proposed reprojection for fully connected layers but still inside the one for convolutional layers. In particular, the top and bottom graphs present the case of negative real stable eigenvalues with different magnitude. The left figures shows that as a result the stable tanh activated networks have monotonic activations (not strictly increasing) that still tend to a straight line as expected but present intermediate decaying oscillations. As a consequence of this the map is still Lipschitz but the Lipschitz constant is not equal to the steady state gain for any $k$. Figures on the right present the ReLU case where the maps remain identical independently of the parameter $A$.

Figure 4: **Single neuron NAIS-Net with variation stopping criteria for pattern-dependent processing depth. Input-output map for different unroll length for tanh (Left) and ReLU (Right) activations.** The input-output maps produced by stable NAIS-Net with our proposed reprojection for fully connected architectures. In particular, the top and bottom graphs present the case of positive stable eigenvalues with different magnitude. The left figures shows that the stable tanh activated networks have piece-wise continuous and locally monotonic activations (strictly increasing around the origin) that tend to a straight line as $k \to \infty$ as in our theoretical results. Moreover, the map is also piece-wise Lipschitz with Lipschitz constant less than the steady state gain presented in the main paper, $\|A^{-1}\| \|B\|$. Figures on the right present the ReLU case where the maps are piece-wise linear functions with slope that is upper bounded by $\|A^{-1}\| \|B\|$. This means that one cannot have an unbounded slope (except for the jumps) or the same map for different unroll length. The rate of change of the map changes as a function of $k$ is also determined by the parameters. The jump magnitude and the slopes are also dependant on the choice of threshold for the stopping criteria.

(a) frog

(b) bird

(c) ship

(d) airplane

Figure 5: **Image samples with corresponding NAIS-Net depth.** The figure shows samples from CIFAR-10 grouped by final network depth, for four different classes. The qualitative differences evident in images inducing different final depths indicate that NAIS-Net adapts processing systematically according characteristics of the data. For example, *"frog"* images with textured background are processed with fewer iterations than those with plain background. Similarly, *"ship"* and *"airplane"* images having a predominantly blue color are processed with lower depth than those that are grey/white, and *"bird"* images are grouped roughly according to bird size with larger species such as ostriches and turkeys being classified with greater processing depth.

## Footnotes

[3] The concept of controllability is not introduced here. In the case of deep networks we just need $B$ to be non-zero to provide input skip connections. For the general case of time-series identification and control, please refer to the definitions in [6].

[4]Here we will consider only the simple case of $y(k) = x(k)$, therefore we can simply use notions of Input-to-State Stability (ISS).

[5] Note that, when using $\tanh$, in the limit case of $\bar{\sigma}^{'} = 0$, $\mathcal{P} = \mathbb{R}^{n+m}$ the system becomes simply stable. This is a small subtlety in our result for $\tanh$, which could be defined as *almost global* or more formally *valid in*

[6]Note that the matrix $A$ is invertible by construction.

[7]Note that the union of all regions of initial conditions converging to a steady state is however the full $\mathbb{R}^n$. Note also that in our application the initial condition is always set to zero as done in the common RNN setting, however, this is not necessary for stability or any other properties.

[8]This follows from the fact that there is a region of the state-input space where the system becomes autonomous, namely, the (constant) input cannot affect the output any longer.

[9]Note that this also implies that $\rho(I + A) \leq 1 - \epsilon$, by means of the matrix norm identity $\rho(\cdot) \leq \|\cdot\|$. This was however already verified by the Gershgorin Theorem.