[Reviews · NeurIPS 2018]

Reviewer 1



This paper proposes a novel Very Deep Neural Architecture, NAIS-Net, in which non-autonomous blocks are stacked. The paper theoretically demonstrate that the network is Globally Asymptotically Stable, when using the tanh activation function. An empirical demonstration of the stability of the architecture is proposed. Quality: The technical content of the paper is well explained and the related work section clearly compare this work with the existing literature. Clarity: The paper is generally well-written and structured clearly. Originality: The main idea of this paper is very interesting and the results look encouraging. Significance: The model proposed is very interesting and the empirical stability looks very promising. The question of the scalability is indeed the major concern. Minor typo: line 221: the last sentence probably refers to Figure 4 and not Figure 3 line 84: usual -> usually Minor comment: Images in Figure 6 are way too small for a printed version...

Reviewer 2



Summarization: This paper presents a deep network that is stable and robust against small perturbations. The model is called Non-autonomous input-output Stable Network (NAIS-Net). The network has two characteristics: the first one is non-autonomous in the sense that skip connections are implemented for each step of the unrolled block. The second characteristic is time-invariance because of the tied weights. The proposed model is able to achieve an input-dependent stable state for the output as well as reduce the generalization gap. The stableness can be enforced by special design of the network parameters. In the end, the authors demonstrate their claims by supportive experiments conducted on MNIST and CIFAR datasets. Strength: 1. The explanatory descriptions and technical details make the work worth of a publication. It is worth to mention that the authors state the motivations clearly by comparing their model with existing works and are able to guide the way directly to the core of their method. Explaining all the necessary technical details and selecting the conclusive ones in the main text without confusing the reader is also a plus. 2. The paper is well structured and explains clearly the background and contributions together with the solutions. Proposals for both fully-connected and convolutional layers are presented. 3. The work presented has its own originality in terms of both model architecture and stability solutions, which makes it a valuable contribution. 4. The authors conduct extensive experiments to support their claims, which include both quantitative and qualitative illustrations. Firgure 3 and 4 demonstrate the convergence stability of the propose method. Specially, Figure 3 shows the effect of each component clearly by comparing different model settings which is convincing. Weakness: 1. For the technical details, although the authors provide sufficient information in the supplementary material, it's still better to state some of them clearly in the main text. E.g., can you explain more on set P (what it represents? Does it include all the pairs that do not saturate? Why the element (x,u) belongs to R^{n*m}, according to supplementary material?), the function of state \hat{x} for both tanh and Relu activations, etc. 2. It would be better to explain more on general stability analysis, the motivation, intuition, what condition is deemed as stable, etc, in order to have an idea on what you are trying to solve. 3. In the beginning, the authors compare their model with ResNet and existing dynamical systems to emphasize the non-autonomous characteristics. However, it lacks some mathematical comparisons. Specifically, why is your method better than ResNet in terms of stability? How is ResNet's formulation hard to achieve stability? 4. The authors may want to explain the what the dotted curve represents in Figure 3. From the method introduction and experiment section, it's not clear to me the effect of time-invariant construction, e.g., compare with ResNet-NA-Stable. 5. Since the model should benefit from better generalization gap, a direct proof could be injecting some noise and compare your model with ResNet. I wonder if you have done this experiment and how's the results? It seems in Table 1, your model does not outperform ResNet. Can you explain possible reasons? And any quantitative results for patter-dependent processing depth?

Reviewer 3



This paper presents NAIS-NET, a model of deep networks constructed from non-autonomous differential equations. The model is a cascade of blocks, each defined by a dynamic state system that is similar to the extended Kalman filter whose state is directly observed. The key differences from ResNet include the control variables u and the re-use of (A, B) parameters within a block. A major advantage of NAIS-NET is that control theory can be applied to enforce stability of each block so that a cascade of blocks can be trained efficiently without batch normalization. Experimental results show that the NAIS-NET has improved generalization gap compared to ResNet. The paper lacks a description of how the NAIS-NET is trained. The presentation can be improved.